# Learning Self-Interpretation from Interpretability Artifacts: Training Lightweight Adapters on Vector-Label Pairs

Keenan Pepper [1]  Alex McKenzie [1]  Florin Pop [1]  Stijn Servaes [1]  Martin Leitgab [1]  Mike Vaiana [1]
Judd Rosenblatt [1]  Michael S. A. Graziano [2]  Diogo de Lucena [1]

## Abstract

Self-interpretation methods prompt language models to describe their own internal states, but remain unreliable due to hyperparameter sensitivity. We show that training lightweight adapters on interpretability artifacts, while keeping the LM entirely frozen, yields reliable self-interpretation across tasks and model families. A scalar affine adapter with just $d_{\text{model}} + 1$ parameters suffices: trained adapters generate sparse autoencoder feature labels that outperform the training labels themselves (70% vs 50% generation scoring at 70B scale), identify topics with 94% recall@1 versus 1% for untrained baselines, and decode bridge entities in multi-hop reasoning that appear in neither prompt nor response, surfacing implicit reasoning without chain-of-thought. The learned bias vector alone accounts for 85% of improvement, and simpler adapters generalize better than more expressive alternatives. Controlling for model knowledge via prompted descriptions, we find self-interpretation gains outpace capability gains from 7B to 72B parameters. Our results demonstrate that self-interpretation improves with scale, without modifying the model being interpreted.

## 1. Introduction

Large language models (LLMs) compute through high-dimensional hidden activations, yet understanding what these internal representations encode remains a central challenge for interpretability and model auditing. Despite substantial progress in mechanistic interpretability (Cunningham et al., 2023; Zou et al., 2023), which has produced large collections of labeled directions in activation space, models cannot straightforwardly report on their own internal states.

A promising direction, *self-interpretation*, uses activation-patching methods to inject internal representations into prompts and generate natural-language explanations (Chen et al., 2024; Ghandeharioun et al., 2024). The appeal is intuitive: the same model that produces a representation may also be well positioned to interpret it, without requiring a separate probe for each concept of interest. In practice, however, these methods are fragile. Small changes in activation scale can produce fluent but semantically ungrounded explanations (Kharlapenko et al., 2024), limiting their reliability as general-purpose interpretability tools.

Recent work improves self-interpretation by fine-tuning language models to answer questions about patched activations (Pan et al., 2024; Li et al., 2025; Karvonen et al., 2025). While effective, these approaches alter the model being interpreted, potentially changing the representations under study. Ideally, the interpreter and the subject model would remain identical.

We train a lightweight adapter that maps internal activations into a form the model can consistently interpret, while keeping the language model completely frozen. We treat existing interpretability artifacts, such as sparse autoencoder (SAE) features paired with labels, or contrastive activation vectors paired with topic descriptions, as supervision. These artifacts provide natural (vector, label) training pairs for learning an activation-to-language mapping.

Across tasks and model families, adapters substantially improve self-interpretation. On topic identification from contrastive activation vectors, recall@1 improves from ~1% to over 90%. On SAE feature labeling, adapters outperform the original training labels under generation-based evaluation (70% vs. 50% at 70B scale). Adapters also recover latent intermediate concepts in multi-hop reasoning tasks, surfacing bridge entities that appear in neither the prompt nor the final response. Because the interpreter is the model itself, these capabilities improve naturally with scale.

Surprisingly, a scalar affine adapter with only $d_{\text{model}} + 1$ parameters achieves most of the gains. The learned bias

[1]AE Studio [2]Princeton Neuroscience Institute & Department of Psychology, Princeton University, Princeton, NJ. Correspondence to: Keenan Pepper <keenan@ae.studio>.

*Proceedings of the $43^{rd}$ International Conference on Machine Learning*, Seoul, South Korea. PMLR 306, 2026. Copyright 2026 by the author(s).

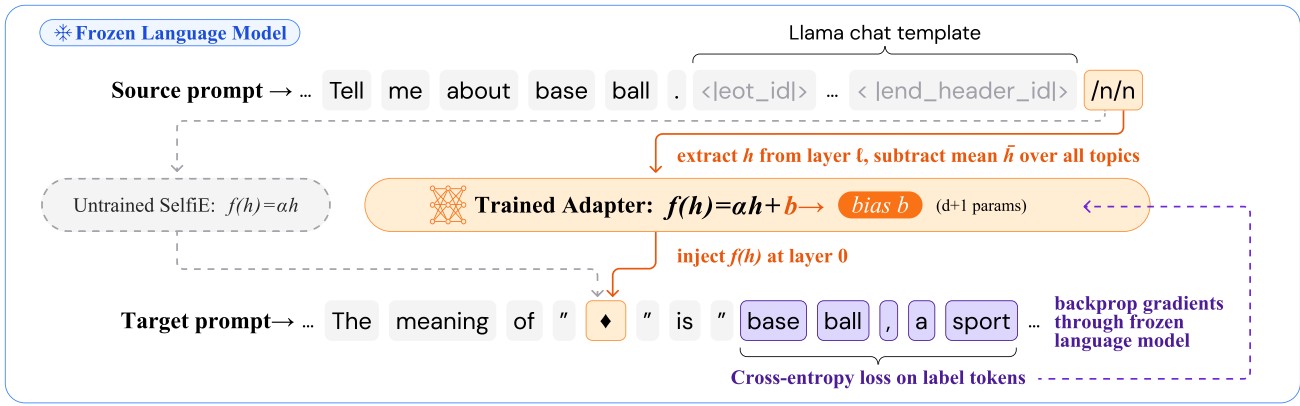

*Figure 1.* **Training self-interpretation from interpretability artifacts.** In this case, the "interpretability artifact" consists of $(h, y)$ pairs where the vector $h$ is a contrastive activation vector from the *source prompt* about a specific topic, and the label $y$ is one of several synthetic descriptions of that topic. The activation $h$ is extracted from layer $\ell$ at the final token position of the source prompt (the \n\n following the chat template's assistant header). A lightweight adapter transforms $h$ and injects it at layer 0 at the placeholder position of an explanation-seeking *target prompt*. Cross-entropy loss on the label tokens trains only the $d_{\text{model}}+1$ adapter parameters; the language model remains frozen. The case of training on an SAE dataset is similar except that $h$ is an SAE decoder vector and $y$ is a natural language feature label, *e.g.* from automated interpretability.

vector alone accounts for roughly 85% of the loss improvement over untrained baselines, acting as an interpretation prior, while the activation vector provides instance-specific semantics. Simpler adapters also generalize better than higher-capacity alternatives across datasets and layers. Together, these results show that reliable self-interpretation can emerge from lightweight transformations without modifying the underlying model.

## 2. Methods

### 2.1. Background: Self-Interpretation via Patching

Self-interpretation methods attempt to decode the semantic content of internal activations by reinserting them into the language model in a generation context. The core idea is to treat an activation vector as a soft prompt, i.e., if the vector encodes a coherent concept, then injecting it into an explanation-seeking prompt may cause the model to describe that concept in natural language.

We adopt the Patchscopes framework (Ghandeharioun et al., 2024): an activation $h_i^\ell$ extracted from layer $\ell$ at position $i$ is transformed via mapping function $f$ and injected into a target prompt $T$ at layer $\ell^*$ and position $i^*$. Following Kharlapenko et al. (2024), we use an explanation-seeking template for $T$:

> **User:**
> What is the meaning of "TOKEN"?
> **Assistant:**
> The meaning of "TOKEN" is "

injecting the transformed activation at the placeholder po-

sition at the token embedding layer $\ell^* = 0$. The injected vector replaces the embedding of the placeholder token at that position.

### 2.2. Trained Adapters

Untrained SelfIE (Self-Interpretation of Embeddings) uses $f(h) = \alpha \cdot h$, but the optimal $\alpha$ varies across vectors and most have narrow valid ranges (Kharlapenko et al., 2024). Rather than manually tuning injection scales for each vector, we learn a transformation $f(h)$ from interpretability datasets consisting of vector-label pairs. We consider adapters with increasing expressivity:

- **Identity**: $f(h) = h$             (0 parameters)

- **Scale-only**: $f(h) = \alpha \cdot h$         (1 parameter)

- **Scalar affine**: $f(h) = \alpha \cdot h + b$    ($d+1$ parameters)

- **Scalar affine + low-rank**: $f(h) = \alpha \cdot h + UV^\top h + b$, where $U, V \in \mathbb{R}^{d \times r}$       ($d+1+2dr$ parameters)

- **Low-rank only**: $f(h) = UV^\top h + b$                   ($d+2dr$ parameters)

- **Full-rank affine**: $f(h) = Wh + b$                       ($d^2+d$ parameters)

Scalar affine adapters treat all directions identically through uniform scaling and recentering, leaving little direction-specific capacity for memorization. Adding a low-rank term ($UV^\top h$) introduces limited direction-specific fitting while $\alpha h$ preserves most directions, whereas full-rank variants can transform every direction arbitrarily.

## 2.3. Training Data: Interpretability Artifacts

We train on vector-label pairs $(h, y)$ from two sources: SAE decoder vectors paired with auto-interpretability labels, and contrastive activation vectors paired with synthetic topic descriptions. SAE features are sparse and approximately monosemantic, while contrastive vectors encode the semantics of a conversation topic by construction. For SAEs we use decoder rather than encoder vectors (Appendix A.1). All input vectors are normalized to unit L2 norm; the adapter learns to map *directions* to descriptions, with a separate scale factor controlling injection magnitude at inference. Dataset details appear in Section 3.1.

## 2.4. Inference Procedure

At inference time, the trained adapter maps an activation vector into the token embedding space and injects the transformed vector into an explanation-seeking prompt. Given an activation $h$ from layer $\ell$, we normalize to $\hat{h} = h/\|h\|_2$, multiply by injection scale $h_s = s \cdot \hat{h}$, apply the adapter to obtain $z = f(h_s)$, inject $z$ at the placeholder token position at layer 0, and generate a description autoregressively.

Because optimal injection magnitudes vary across vectors and datasets, we evaluate multiple scales per vector and select among the resulting candidate generations. The adapter is trained once and reused across inference tasks; no sparse autoencoder is required during inference beyond the source activation vector itself.

## 2.5. Training Objective

We minimize cross-entropy loss averaged over label tokens (standard supervised learning with no autoregressive generation during training). Given $(h, y)$, we inject $f(h)$ at the placeholder position and compute loss against $y$ (see Appendix A.1 for label formatting details). The language model remains frozen; only adapter parameters are trained.

## 2.6. Evaluation and Scale Selection

The injection scale $s$ is drawn from a fixed logarithmic grid calibrated once per adapter-dataset pair (Appendix A.5). We generate $N=6$ candidate descriptions per vector by varying the injection scale, giving each method an equal candidate budget. To avoid selection-on-noise effects, candidate selection and final reporting use independent scoring seeds. Training uses known vector-label pairs from interpretability datasets, while inference applies the learned adapter to previously unseen activations.

## 3. Experiments

We evaluate trained adapters across two representation classes (three datasets) that differ substantially in geom-

etry: contrastive activation vectors, and sparse autoencoder feature directions. §3.1 establishes models, datasets, and evaluation protocols. §3.2 shows that full-rank adapters recover semantic topics from contrastive vectors with high accuracy, raising the question of whether the same approach transfers to SAE features. §3.3 answers this by analyzing adapter architecture and representation geometry, identifying why full-rank succeeds in one setting but overfits in the other. §3.4 evaluates these architectures on cross-dataset generalization and larger models. §3.5 examines how self-interpretation scales with model size relative to a capability ceiling. §3.6 tests whether adapters trained on monosemantic vectors generalize to arbitrary residual stream activations, using implicit multi-hop reasoning as a testbed.

### 3.1. Experimental Setup

**Models.** Primary experiments use Llama-3.1-8B-Instruct, with additional validation on Llama-3.3-70B-Instruct and Gemma-2-9B-IT. For scaling analysis we use the Qwen-2.5-Instruct family (7B, 14B, 32B, 72B). Cross-family transfer results appear in Appendix G.

**Datasets.** We train and evaluate on two classes of vector-label pairs across three datasets. The first consists of SAE feature directions paired with auto-interpretability labels. We use two SAE families:

(1) *Goodfire SAE features:* 45,418 decoder vectors from a layer 19 residual stream SAE on Llama-3.1-8B-Instruct (Balsam et al., 2025), paired with auto-interpretability labels. For Llama-3.3-70B-Instruct, we use 61,521 features from the layer 50 SAE.

(2) *Llama Scope SAE features:* decoder vectors from 32k- and 131k-width SAEs trained on layers 0–31 of Llama-3.1-8B (base) (He et al., 2024), with auto-interpretability labels available on Neuronpedia. Despite being trained on the base model, these SAEs transfer well to the instruction-tuned variant (Kissane et al., 2024).

The second class consists of semantic directions from contrastive prompting:

(3) *Wikipedia contrastive vectors:* Following Lindsey (2025), we compute activations for prompts of the form "Tell me about [topic]" at layer 19 (chosen to match the Goodfire SAE layer) and subtract the mean activation to obtain contrast vectors for 49,637 Wikipedia "Vital Level 5" article titles. Each vector is paired with approximately 15–20 synthetic labels describing the topic at varying levels of detail, including synonyms and paraphrases.

**Evaluation.** We evaluate whether the adapter-mediated self-interpretation pipeline faithfully recovers the semantics encoded in an activation vector.

For SAE features, we use two complementary evaluations. Detection scoring (Paulo et al., 2024) tests whether a generated description correctly recognizes when a feature is active on held-out contexts. Generation scoring (Juang et al., 2024) tests whether a description can recreate activating contexts from scratch: given a generated label, we prompt the model to produce matching text and measure whether the corresponding feature activates on those outputs. The two metrics can disagree in practice (see Appendix K). A broad label may recognize many activating contexts while generating weakly, whereas a precise label may strongly recreate the feature while missing some valid contexts. We report hit rate (the percentage of generations with at least one nonzero activation) and coverage (the percentage of latents ever receiving a nonzero activation). Generation scoring is particularly relevant for self-interpretation because the task begins from an activation vector rather than a dataset of activating examples.

For contrastive vectors, we evaluate via embedding-based retrieval. Each topic is represented as a document embedding (title plus all labels) using GTE-large (Li et al., 2023). We embed each generated description and report recall@$k$, the fraction of topics for which the correct topic appears among the top-$k$ nearest neighbors.

## 3.2. Contrastive Activation Vector Results

We train a full-rank affine adapter $f(h) = Wh + b$ for one epoch on the Wikipedia contrastive vectors dataset. Without scale tuning (using the as-trained injection scale $s = 1$), the full-rank adapter achieves 82.9% recall@1 on held-out topics compared to 0.04% for untrained SelfIE, and 98.4% versus 0.9% at recall@100. Note that these results use only the trained adapter, without additional input scaling or best-of-N selection which provide further improvement (discussed in Table 2). The full-rank adapter outperformed constrained architectures (Appendix D). As §3.3 shows, this success depends critically on the low intrinsic dimensionality of the contrastive topic vectors, which implicitly regularizes the full-rank transformation.

**Qualitative example.** To illustrate that trained adapters extract semantic concepts rather than surface-level token information, we apply the method to a novel prompt (absent from the dataset): "Tell me about propagating gradients back through a neural network." Five generations at temperature 0.5 all converge on the same concept:

- "automatic differentiation for backpropagation in neural networks"

- "backpropagation in neural network training"

- "automatic differentiation in deep learning and computational science"

- "automatic differentiation for neural network backpropagation"

- "backpropagation in neural networks"

The adapter consistently identifies the core concept despite the prompt never using the word "backpropagation." Our next analysis asks whether this approach transfers to SAE features, which occupy a geometrically different region of activation space.

## 3.3. Adapter Architecture and Representation Geometry

The full-rank approach that succeeded on contrastive vectors is not optimal for SAE features, because it overfits, and early stopping does not fully mitigate this. We therefore begin by characterizing which adapter architectures work for this more challenging setting. Table 1 compares adapter architectures trained on Llama Scope SAE features (see Appendix G for replication on Gemma Scope).

*Table 1.* Adapter architecture comparison on Llama Scope SAEs (layer 19, 32k and 131k widths combined). SA = scalar affine, LR = low-rank. Parameter counts shown for Llama-3.1-8B ($d$=4096). The bias vector accounts for the majority of improvement; adding LR yields further gains while full-rank dramatically overfits.

| ARCHITECTURE | PARAMS | VAL LOSS | Δ |
|---|---|---|---|
| IDENTITY | 0 | 4.834 | — |
| SCALE-ONLY | 1 | 4.543 | -0.291 |
| SCALAR AFFINE | 4097 | 1.787 | -3.047 |
| SA + LR (R=4) | 37K | 1.694 | -3.140 |
| SA + LR (R=16) | 135K | 1.637 | -3.197 |
| SA + LR (R=64) | 528K | **1.619** | **-3.215** |
| SA + LR (R=256) | 2.1M | 1.622 | -3.212 |
| LR ONLY (R=4) | 37K | 2.002 | -2.832 |
| LR ONLY (R=16) | 135K | 1.766 | -3.068 |
| LR ONLY (R=64) | 528K | 1.646 | -3.188 |
| LR ONLY (R=256) | 2.1M | 1.642 | -3.192 |
| FULL-RANK AFFINE (TRAIN LOSS) | 16.8M | 1.743 (0.64) | -3.091 |

**The bias vector is critical.** Scale-only improves just 0.29 over identity, but adding the bias vector (scalar affine) yields an additional 2.75 improvement. This single $d$-dimensional vector accounts for approximately 85% of the total gain from our best adapter.

**Scalar affine is a strong minimal baseline.** With only $d+1$ parameters, scalar affine achieves validation loss of 1.787, already a large improvement over untrained approaches with essentially zero train-val gap.

*Table 2.* Cross-dataset generalization. Each method produces 6 candidate labels per held-out latent or topic (adapters and untrained SelfIE vary scale; SAE baselines use either 6 copies of the auto-interp label or the original plus 5 LLM paraphrases). See Appendix A.5 for scoring procedure details. "Hit rate" measures percent of generations with at least one nonzero activation; "Coverage" measures percent of latents ever receiving a nonzero activation; "Det. F1" is from `delphi` detection scoring. Subscripts show SEM; bold indicates no significant difference from the best model in each column (paired t-test, $p \geq 0.05$).

| | LLAMA SCOPE SAEs | | | GOODFIRE SAEs | | WIKIPEDIA TOPICS | |
| METHOD | HIT RATE | COVERAGE | DET. F1 | HIT RATE | COVERAGE | R@1 | R@100 |
|---|---|---|---|---|---|---|---|
| *Trained on Llama Scope SAE:* | | | | | | | |
| SA+LR | $44.6_{\pm 0.8}$ | $\mathbf{68.6}_{\pm 0.8}$ | $\mathbf{0.722}_{\pm 0.003}$ | $42.1_{\pm 0.6}$ | $76.1_{\pm 0.6}$ | $0.8_{\pm 0.1}$ | $13.4_{\pm 0.5}$ |
| SA | $46.1_{\pm 0.8}$ | $\mathbf{67.8}_{\pm 0.8}$ | $0.706_{\pm 0.003}$ | $52.1_{\pm 0.6}$ | $83.9_{\pm 0.5}$ | $8.7_{\pm 0.4}$ | $44.6_{\pm 0.7}$ |
| *Trained on Goodfire SAE:* | | | | | | | |
| SA+LR | $43.6_{\pm 0.8}$ | $62.5_{\pm 0.9}$ | $0.677_{\pm 0.003}$ | $55.2_{\pm 0.6}$ | $87.4_{\pm 0.5}$ | $2.8_{\pm 0.2}$ | $19.9_{\pm 0.6}$ |
| SA | $\mathbf{47.6}_{\pm 0.8}$ | $67.1_{\pm 0.8}$ | $0.677_{\pm 0.003}$ | $\mathbf{59.2}_{\pm 0.6}$ | $87.7_{\pm 0.5}$ | $8.4_{\pm 0.4}$ | $42.9_{\pm 0.7}$ |
| *Trained on Wikipedia topics:* | | | | | | | |
| FULL-RANK | $22.1_{\pm 0.7}$ | $36.1_{\pm 0.8}$ | $0.362_{\pm 0.004}$ | $32.0_{\pm 0.6}$ | $60.7_{\pm 0.7}$ | $\mathbf{93.7}_{\pm 0.3}$ | $\mathbf{99.6}_{\pm 0.1}$ |
| SA | $39.3_{\pm 0.8}$ | $56.4_{\pm 0.9}$ | $0.519_{\pm 0.004}$ | $47.0_{\pm 0.6}$ | $74.0_{\pm 0.7}$ | $79.4_{\pm 0.6}$ | $97.9_{\pm 0.2}$ |
| *Baselines:* | | | | | | | |
| UNTRAINED SELFIE | $30.1_{\pm 0.7}$ | $48.8_{\pm 0.9}$ | $0.577_{\pm 0.004}$ | $36.4_{\pm 0.6}$ | $69.1_{\pm 0.7}$ | $1.3_{\pm 0.2}$ | $17.2_{\pm 0.5}$ |
| AUTO-INTERP LABELS | $36.6_{\pm 0.7}$ | $65.5_{\pm 0.8}$ | $\mathbf{0.722}_{\pm 0.003}$ | $50.8_{\pm 0.6}$ | $87.2_{\pm 0.5}$ | — | — |
| ORIGINAL + 5 PARAPHRASES | $41.1_{\pm 0.7}$ | $\mathbf{67.8}_{\pm 0.8}$ | $0.712_{\pm 0.003}$ | $56.1_{\pm 0.6}$ | $\mathbf{89.5}_{\pm 0.5}$ | — | — |

**Low-rank additions provide meaningful gains.** Adding a low-rank component on top of scalar affine yields consistent improvements, with rank 64 achieving the best validation loss of 1.62. Returns diminish beyond rank 64.

**The identity structure matters.** Comparing low-rank only versus scalar affine + low-rank at matched ranks reveals a consistent gap favoring the identity-preserving variant. This gap is most dramatic at low rank: at rank 4, adding the scalar affine base improves validation loss from 2.00 to 1.69, a difference of 0.31, larger than the entire gain from increasing rank. The scaled identity term preserves directional information that pure low-rank transformations discard.

**Full-rank transformations overfit catastrophically on SAE data.** Full-rank adapters have $4096\times$ more parameters than scalar affine, yet achieve similar or worse validation loss while reaching train loss of 0.64. Early stopping cannot help: SA+LR already outperforms full-rank's best point (1.66 vs 1.70) after just one epoch, and continues improving to 1.62 by epoch 3. Full-rank transformations overfit on SAE data by effectively learning a high-dimensional lookup table, destroying identity-preserving structure in the activation space. See Appendix B for training curves.

This contrast traces to intrinsic dimensionality. Wikipedia contrastive vectors concentrate over 90% of variance in $\sim$200 dimensions, providing implicit regularization for full-rank transformations, while SAE features span nearly the full activation space. Controlled experiments ruling out label count as a confound appear in Appendix D. Scale selection behavior is detailed in Appendix A.5. We apply

these findings to evaluate performance across datasets and model scales in §3.4.

### 3.4. Cross-Dataset Generalization and Evaluation

Table 2 shows generalization across datasets and evaluation types. Each row shows an adapter (or baseline) evaluated on all three evaluation settings; blocks on the diagonal represent in-distribution performance (trained and evaluated on the same dataset). Scalar affine adapters generalize better across datasets than higher-capacity adapters. The Wikipedia SA adapter achieves 39.3% and 47.0% generation scoring hit rates on SAE latents despite never seeing SAE features during training, while the Wikipedia full-rank adapter (optimal in-distribution) achieves only 22.1% and 32.0%. This pattern replicates on Gemma-2-9B-IT (Appendix G). Second, Goodfire-trained adapters achieve the highest generation scores even on Llama Scope evaluation, suggesting the Goodfire labels may be higher quality or better suited to instruction-tuned models.

The detection scoring column provides an independent validation using a standard SAE evaluation metric (Paulo et al., 2024). The SA+LR adapter trained on Llama Scope achieves the highest detection F1 (0.722), tied with the auto-interp baseline (also 0.722) and slightly above the auto-interp paraphrases (0.712). The trained adapter therefore matches, but does not exceed, the strongest baseline on detection scoring, while substantially outperforming all baselines on generation scoring (the Goodfire-trained SA adapter reaches 47.6% hit rate on Llama Scope versus 41.1% for the strongest baseline). The best-performing architecture also

differs across the two SAE scoring protocols: SA+LR ties on detection while SA wins on generation. Detection and generation scoring measure related but distinct properties of a label (recognition vs. specification), so the labels that maximize one metric are not necessarily those that maximize the other.

**Comparison with LoRA fine-tuning.** We trained LoRA-based Activation Oracles (Karvonen et al., 2025) of rank 4 on the same datasets with the same evaluation protocol. On Wikipedia topics, LoRA achieves $81.6\%_{\pm0.6}$ recall@1 versus $82.9\%_{\pm0.6}$ for trained SelfIE. On Goodfire SAE generation scoring, LoRA achieves $62.3\%_{\pm0.6}$ hit rate versus $59.2\%_{\pm0.6}$ for trained SelfIE. Performance is comparable; however, trained SelfIE leaves model weights unchanged.

*Table 3.* SAE label evaluation on Llama-3.3-70B-Instruct on held-out Goodfire SAE latents. Each method produces 6 candidate labels per latent (SelfIE methods vary scale; baselines use either 6 copies of the auto-interp label or the original plus 5 LLM paraphrases). See Appendix A.5 for scoring details. The trained adapters decisively outperform both untrained SelfIE and the training labels on both metrics.

| METHOD | HIT RATE | COVERAGE | DET. F1 |
|---|---|---|---|
| ADAPTER (SA) | $66.8_{\pm1.3}$ | $\mathbf{83.2}_{\pm1.2}$ | $\mathbf{0.760}_{\pm0.007}$ |
| ADAPTER (SA+LR) | $\mathbf{69.7}_{\pm1.3}$ | $82.2_{\pm1.2}$ | $0.747_{\pm0.007}$ |
| ORIG. + 5 PARAPHRASES | $60.4_{\pm1.4}$ | $79.8_{\pm1.3}$ | $0.701_{\pm0.008}$ |
| AUTO-INTERP $\times6$ | $50.0_{\pm1.4}$ | $71.5_{\pm1.4}$ | $0.673_{\pm0.008}$ |
| UNTRAINED SELFIE | $48.1_{\pm1.5}$ | $61.9_{\pm1.5}$ | $0.538_{\pm0.011}$ |

**Evaluation on Llama-3.3-70B.** In self-interpretation, scaling the subject model simultaneously scales the interpreter, since they are the same model. Table 3 shows results on Llama-3.3-70B-Instruct, where we evaluate trained adapters on held-out latents from a Goodfire SAE (layer 50, 61k features).

The trained SA+LR adapter achieves 69.7% mean hit rate (66.8% for the simpler SA adapter), decisively outperforming both untrained SelfIE (48.1%) and the training labels themselves, whether repeated six times (50.0%) or paraphrased (60.4%). The adapter also achieves higher coverage: 83.2% of latents receive at least one accurate label (nonzero hit rate) compared to 79.8% for paraphrases and 61.9% for untrained SelfIE. Detection scoring shows the same ordering: both trained adapters (F1 = 0.760 for SA, 0.747 for SA+LR) outperform paraphrases (0.701), repeated auto-interp labels (0.673), and untrained SelfIE (0.538). At the 70B scale, the SA and SA+LR adapters trade off across the two metrics: SA wins on detection here while SA+LR wins on generation.

These results demonstrate that trained self-interpretation scales favorably with model size. The 70B model both

*Table 4.* Greedy-decoded labels for held-out Goodfire SAE latent #41101. The training label is generic, shared by 51 latents. Untrained SelfIE produces confident but ungrounded descriptions that vary erratically with scale; the trained adapter produces accurate, consistent descriptions. **Score**: generation scoring hit rate (percentage of synthetic texts matching the label description that elicit nonzero activation from this latent)

| Label | Score |
|---|---|
| *Training label (auto-interpretability):* | |
| "The assistant should complete a code snippet" | 2% |
| *Untrained SelfIE:* | |
| [scale 1.0] "a small hill or mound of earth" or "a small hill or ridge of land". It can also refer to a small hill or mound of earth that is used as a burial site... | 0% |
| [scale 2.0] "a device or system that detects and reports a specific event or condition, typically used in computer science and electronics." In other words, a "notification" or an "alarm" that is triggered... | 17% |
| [scale 3.0] "a formal meeting or gathering, especially one for a special purpose" or "a formal or official announcement or statement, especially one made in a public place."... | 3% |
| *Trained adapter (SA + LR, r=16):* | |
| [scale 1.0] "Event listener and callback function definitions in code" | 100% |
| [scale 2.0] "The code that defines a new event handler or callback function" | 98% |
| [scale 3.0] "The code that defines the event handling function, typically in a GUI or event-driven application" | 100% |

requires more sophisticated interpretation (larger embedding space, more complex representations) and provides it (more capable language generation).

**Qualitative example.** Table 4 illustrates how trained adapters can outperform both untrained SelfIE and the training labels themselves. This held-out Goodfire latent has a uselessly generic training label. Untrained SelfIE produces fluent but erratic descriptions; the trained adapter consistently produces accurate descriptions with 98–100% hit rates, confirming they faithfully characterize when the latent fires.

This example is cherry-picked but not unique. In the 8B Goodfire SAE, 30 latents (out of 4541 in the validation set) had a maximum hit rate of 0/10 across all auto-interp label paraphrases, yet some SelfIE-generated labels achieved 10/10. For the Llama Scope SAE, 87 latents (out of 3243) showed this same "extreme improvement."

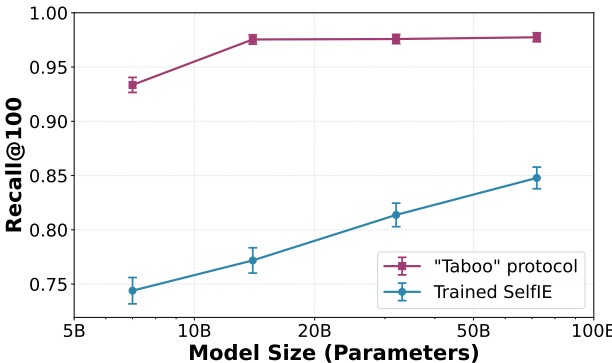

*Figure 2.* Scaling comparison on Qwen-2.5 models (7B to 72B). **Trained SelfIE** (below): recall@100 on held-out topics for full-rank adapters trained on contrastive topic vectors from the middle half of each model's layers. **Taboo baseline** (above): the model describes each topic without naming it, scored with the same embedding retrieval. While SelfIE consistently performs below the Taboo ceiling, the gap decreases with model scale as SelfIE's performance increases more rapidly. Error bars show 95% confidence intervals. See Appendix F for additional metrics.

### 3.5. Self-Interpretation Scales with Model Size

Larger models generally perform better at nearly every task, so showing that self-interpretation improves with scale would prove little on its own. To isolate self-interpretation gains from general capability improvements, we compare against a "Taboo" baseline: the model describes each topic without mentioning its name, scored with the same embedding-based retrieval. This measures how much topic knowledge the model can express directly, providing a ceiling for what self-interpretation could hope to extract.

We train full-rank adapters on contrastive activation vectors from Qwen-2.5 models at four scales (7B, 14B, 32B, 72B). For each model, we pool vectors from the middle half of all layers (e.g., layers 7–20 for a 28-layer model), training a single adapter that receives no layer index yet interprets activations from any training layer. This cross-layer generalization replicates findings from Llama (Appendix E).

Figure 2 shows the results. The model's underlying knowledge of the topics saturates at an intermediate scale, but the performance of trained SelfIE grows faster, revealing more of the semantic information that's present. Untrained SelfIE achieves <2% recall@100 at all sizes. Unlike Table 2, these results use only the trained adapter with no additional input scaling and no best-of-N protocol.

### 3.6. Application: Decoding Implicit Reasoning

The preceding experiments use vectors that are approximately monosemantic, but the broader promise of self-interpretation is reading out *arbitrary* internal states, including polysemantic activations. We test whether adapters

trained on monosemantic vectors generalize to this more challenging setting, particularly, multi-hop reasoning, where models must implicitly represent intermediate conclusions. Using prompts from TwoHopFact (Yang et al., 2024), we demonstrate that even when the model responds immediately with the correct answer, with no verbalized chain of thought (CoT), our method can often extract the *bridge entity* that appears in neither prompt nor response.

Consider "The author of the novel The Republic was born in the city of." Llama3.1-8B-Instruct correctly responds "Athens," with no mention of the bridge entity "Plato." Did the model shortcut directly to the answer, or did it construct a latent representation of "Plato" that was never verbalized?

We filter TwoHopFact for questions such that (1) the model can correctly name the bridge entity when asked the first-hop question *and* (2) the model can correctly answer the two-hop question immediately, with no verbalized CoT. This yielded 5546 questions (12% of the dataset). We then sample 500 of these, and generate SelfIE descriptions (untrained and trained) at every layer and token.

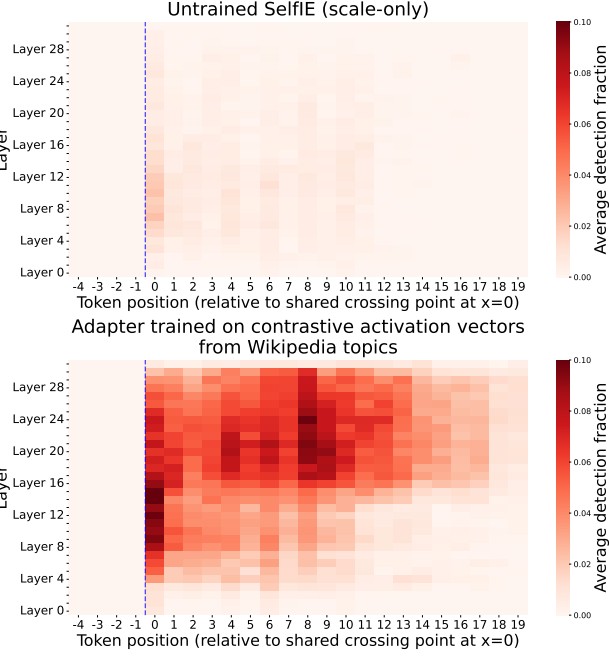

*Figure 3.* Bridge entity detection across layers and token positions. Each cell shows the fraction of SelfIE generations (temperature 0.7, 10 samples per cell) containing any alias of the bridge entity (e.g., "Plato" for the prompt "The author of The Republic was born in the city of"). Position 0 is aligned to the first token where detection exceeds 0.1%; negative positions are earlier context. **Top:** Untrained SelfIE shows weak, localized signal. **Bottom:** Trained adapter (scalar affine) produces stronger detection rates over a broader range of layers and positions. Aggregated over 500 randomly sampled TwoHopFact prompts where the language model answers both two-hop and first-hop questions correctly when instructed to answer immediately with no CoT.

Figure 3 shows aggregate heatmaps across token positions and layers. Bridge entity detection typically becomes possible only after a specific token (e.g., before ' Republic' the model cannot infer Plato). We align heatmaps by defining position 0 as the first position where detection crosses a low threshold (0.001) for *either* method.

Untrained SelfIE shows weak signal mostly confined to a single token; trained adapters substantially increase detection rates across a broader range. Across all 500 prompts, the trained adapter detected the bridge entity in 455 cases ($91.0\%_{\pm1.3}$) versus 282 ($56.4\%_{\pm2.2}$) for untrained; this is a $4.8\times$ reduction in the number of undetected bridge entities. Only 2/500 prompts showed the reverse pattern (untrained success, trained failure), indicating near-zero regression. For questions where both methods succeeded in extracting the bridge entity, the trained adapter does so much more reliably: $4.26\%_{\pm0.24}$ of all generations compared to $0.38\%_{\pm0.04}$ for untrained (an $11\times$ increase). (SAE-trained adapters perform worse at this task than adapters trained on contrastive topic vectors, yet still beat the untrained baseline; Appendix H.)

**Comparison with linear probes.** We compare against linear probes trained to map last-token activations to GTE-large embedding space via InfoNCE loss, one probe per layer. At $n=10$ generations per position, SelfIE achieves 73.0% bridge entity detection at its best layer versus 67.0% for the best probe layer. When pooling across all layers and token positions (32 probes versus one SelfIE adapter), SelfIE achieves $91.0\%_{\pm1.3}$ versus 85.6% for probes at $n=10$. Unlike probes, SelfIE produces free-form natural language descriptions rather than selecting from a fixed entity set. This capability could be useful for decoding what the model is thinking even when verbalized CoT is unavailable or suspect. Amodei (2025) calls this "catching models red-handed," noting that if models can act on representations they never verbalize, surfacing such latent states is crucial for alignment auditing.

## 4. Related Work

**Self-interpretation in language models.** Chen et al. (2024) introduce SelfIE, prompting language models to describe their own activations by injecting them into an explanation-seeking prompt. Ghandeharioun et al. (2024) generalize this with Patchscopes, a unifying framework that casts many prior interpretability methods as instances of activation patching, including Logit Lens (nostalgebraist, 2020), Tuned Lens (Belrose et al., 2023), and causal tracing (Meng et al., 2022). Our work builds on these methods, adopting the Patchscopes formalism and focusing on improving the reliability of self-interpretation by training the mapping function $f$ rather than using an untrained $f$ such as the identity.

**Limitations of untrained self-interpretation.** Kharlapenko et al. (2024) discovered that the magnitude of injected vectors is critical, different vectors require different optimal scales, and most have only narrow ranges producing accurate outputs. Importantly, outputs outside these ranges remain fluent, with the model confidently generating plausible descriptions ungrounded in the vector's semantics. This makes failure modes difficult to detect and motivates learning the mapping function rather than hand-tuning it.

**Learning to interpret activations.** Pan et al. (2024) introduce LatentQA, fine-tuning decoder LLMs via LoRA to answer questions about patched-in activations. This approach frames activation interpretation as question-answering, enabling flexible queries about model internals. Extending this direction, Li et al. (2025) train models to explain feature descriptions, activation patching outcomes, and input ablation effects, finding that models explain themselves better than other models can (the "privileged access hypothesis"). Karvonen et al. (2025) train on diverse tasks including system prompt Q&A, classification, and self-supervised context prediction, achieving strong out-of-distribution generalization to auditing tasks like recovering secrets fine-tuned into models. Huang et al. (2025) train encoder-decoder architectures with sparse bottlenecks, where the encoder compresses activations to interpretable concepts and the decoder answers questions about model behavior.

All of these approaches modify the explainer model's weights. We take a complementary approach: freeze the model entirely and train only a lightweight affine transformation *before* injection. This distinction parallels fine-tuning versus soft prompt optimization (Lester et al., 2021; Li & Liang, 2021): where concurrent work modifies model weights, we modify only the input representation. This preserves "privileged access" maximally—the interpreter is identical to the subject. Fine-tuning moves along a spectrum of decreasing similarity; keeping the model frozen is the most conservative choice for preserving whatever privileged access exists. The two approaches (decoder finetuning and trained adapters) are complementary and could be combined, but we leave this to future work.

**Learned mappings for activation interpretation.** Tuned Lens (Belrose et al., 2023) also learns affine transformations of hidden states, but projects activations *out* of the model to vocabulary space for single-token prediction. Our adapters project activations *back into* the model for open-ended autoregressive generation, a less constrained target that may explain why full-rank can overfit catastrophically for us (on SAE data) while Tuned Lens benefits from it.

# 5. Discussion

**What the adapter learns.** The bias vector encodes a "default interpretation prior": when applied to a zero vector, it generates generic descriptions matching the training distribution (Appendix J). Training on ALL-CAPS labels yields both capitalized and semantically accurate outputs, confirming the bias captures format while the vector contributes semantics (Appendix I). The scale controls how much input semantics override the prior; the scale minimizing cross-entropy often differs from that maximizing generation scoring. This suggests future work on objectives that directly optimize interpretation quality rather than label likelihood.

**Why adapters can surpass training labels.** Auto-interpretability labels are noisy, generated once per feature by prompting an LLM with activation examples. Our adapter learns a mapping function across thousands of examples; if it captures genuine structure in how vectors relate to semantics, it can produce descriptions more accurate than individual noisy labels, just as regression predictions can be more accurate than noisy data points.

**Scaling to frontier models.** The scaling trend in Figure 2 is striking: self-interpretation improves steadily with model size and shows no sign of plateauing at 72B. Training these adapters is computationally cheap ($\sim$10 GPU-hours at 70B scale) and requires only vector-label pairs that frontier labs already produce in abundance (millions of labeled SAE features). For organizations with such datasets, training a self-interpretation adapter is a natural experiment.

**From monosemantic to polysemantic.** Our adapters train on monosemantic vectors (SAE features, contrastive vectors) yet transfer to polysemantic residual stream activations. That this works at all is somewhat surprising. The Q&A paradigm from concurrent work (Pan et al., 2024; Li et al., 2025; Karvonen et al., 2025) suggests an obvious extension: rather than training on a single "what is the meaning of...?" question, train on diverse questions that probe different facets of the activation's semantics. This lets the activation remain polysemantic while still yielding interpretable answers: the question selects which aspect to surface.

**Toward verifiable self-interpretation.** Our work provides infrastructure for training models to accurately report their internal states (what Kim et al. (2025) call "agentic interpretability"). But cooperation is only valuable if trustworthy. Both trained adapters and fine-tuned decoders can learn to produce plausible descriptions ungrounded in activation semantics. They differ in capacity for such shortcuts: scalar affine adapters cannot learn input-dependent patterns, while fine-tuning introduces millions of parameters with more capacity for spurious patterns. Our approach has fewer degrees of freedom for things to go wrong.

Generation scoring makes self-interpretation testable: a model's claim about an internal feature can be checked against behavior. Testable claims can become training signal. We call this direction *RL from internal rewards*: optimizing models to accurately report their own computations, using the same paradigm now driving capability gains. If privileged self-access is real, the best interpreter of a model may be itself, once given opportunity to learn. The long-term vision is models that provide not just fluent self-reports, but verifiable evidence of their own internals.

# 6. Conclusion

We train lightweight adapters on interpretability artifacts to improve self-interpretation. The bias vector accounts for most improvement; scalar affine adapters generalize best across datasets; full-rank overfits on SAE data but succeeds on contrastive topic vectors. Adapters generalize across datasets, layers, and from monosemantic training to polysemantic inference, and our results generalize to other language model families.

The core insight is simple: mechanistic interpretability research has already produced large quantities of structured knowledge about model internals in the form of labeled vectors. Rather than treating these artifacts as endpoints of analysis, we can treat them as training data. This reframing opens a path toward self-interpretation systems that improve automatically as interpretability research progresses.

A complementary approach (Li et al., 2025; Karvonen et al., 2025; Huang et al., 2025) relies on fine-tuning the LLM itself to answer questions about its activations. These methods can recover fine-tuned secrets, detect jailbreaks, and answer arbitrary natural-language queries, at the cost of modifying model weights. Our lightweight adapters offer a different trade-off: fewer parameters, a frozen base model, and the ability to precisely characterize what the transformation learns.

Several directions remain unexplored. Adopting the Q&A framing and diverse training objectives from concurrent work, particularly self-supervised context prediction, could substantially expand our adapters' capabilities while preserving their simplicity. The auditing applications these methods demonstrate (recovering hidden objectives, detecting behavioral changes from fine-tuning) are compelling targets that lightweight adapters might address with appropriate extensions.

## Impact Statement

This work aims to improve the transparency and interpretability of large language models by enabling them to reliably describe their own internal representations. Such capabilities could contribute to AI safety by helping researchers and practitioners detect when models encode concepts or intentions that differ from their expressed outputs. The lightweight, frozen-model approach we take preserves the behavior of the model being interpreted, avoiding concerns that fine-tuning might introduce artifacts that mask the original model's representations.

We note that self-interpretation, even when accurate, provides only partial visibility into model computations. Verification that generated labels match activation patterns does not guarantee coverage of all safety-relevant features, nor robustness to models that might learn to produce misleading self-descriptions. Self-interpretation should complement, not replace, other alignment approaches.

## Acknowledgements

The authors gratefully acknowledge funding from the AI Alignment Foundation in support of this research.

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

# A. Experimental Details

## A.1. Dataset Details for SAE Datasets

We use SAE decoder rather than encoder vectors because decoder vectors represent feature directions in activation space directly, whereas encoder vectors are optimized for sparse reconstruction and may not preserve interpretable structure as faithfully.

Since the SelfIE template ends with an opening double quote (see Section 2.1), we append a closing double quote and end-of-turn token to each label, teaching the model to produce a single complete description and then stop ("teaching" via the influence of the soft token on generation, since the language model parameters are all frozen).

## A.2. Dataset Details for Wikipedia Topics Dataset

Wikipedia's "Vital Level 5" articles comprise approximately 50,000 topics deemed essential for a comprehensive encyclopedia. We compute contrastive activation vectors by running "Tell me about [topic]" prompts through the model, extracting residual stream activations at layer 19, and subtracting the mean activation across all topics.

Each topic is paired with 6–20 synthetic labels generated by Claude Sonnet 4.5, describing the topic at varying levels of specificity and from different angles. The following prompt was used:

```
You are helping create a dataset of
   conversational prompts about Wikipedia
   topics.
```

```
For each Wikipedia article title below,
    first decide on a single meaning the
    article is probably about, e.g. "Bit"
    is about binary bits not screwdriver
    bits. (It's actually okay if you get
    this wrong, as long as you're
    consistent!) Then provide:
1. A natural conversational prompt
    starting with "Tell me about" (e.g.,
    "Tell me about bits (binary digits)."
    or "Tell me about the Riemann
    hypothesis.")
2. Five varied labels for this topic with
    different levels of detail:
    - One medium with brief context (5-10
    words)
    - One with a definition (10-20 words)
    - One with domain context (e.g.,
    "Factorials in combinatorics")
    - At least one which doesn't begin with
    the Wikipedia article title itself
    (e.g. use an alternate name, or start
    with other words)
    - Your choice to fill out the rest of
    the five labels

**CRITICAL: Every label must UNIQUELY
    identify the topic!**
- BAD: "German poet and writer of the 19th
    century" (could be anyone!)
- GOOD: "Heinrich Heine" or "Heinrich
    Heine, German poet of the 19th
    century" or "the author of Die Lorelei"
- BAD: "Chinese fantasy novel depicting
    deification" (too generic!)
- GOOD: "Investiture of the Gods" or
    "Fengshen Yanyi" or "the Ming novel
    Investiture of the Gods"
- BAD: "the tale of how ancient heroes
    became divine" (way too vague!)
- GOOD: "Investiture of the Gods" or "the
    deification narrative in Fengshen
    Yanyi"

Either include the actual title/name, or
    use a specific alternate name/synonym,
    or add enough specificity to make it
    unambiguous.

Format your response as JSON:
{
  "original_title": "Factorial",
  "prompt": "Tell me about factorials.",
  "labels": [
    "factorial",
    "the factorial function in
    mathematics",
    "factorial, the product of all
    positive integers less than or equal
    to n",
    "factorials in combinatorics and
    probability",
    "the n! notation for consecutive
    integer products"
```

```
    ]
}

Be natural with grammar for the prompt:
- Use articles where needed ("the Riemann
    hypothesis")
- Pluralize general concepts ("theorems",
    "buildings")

For the labels:
- Start with lowercase unless the first
    word is a proper noun or always
    capitalized (e.g., "the founder of
    R.K. Films" not "The founder of R.K.
    Films")

Start IMMEDIATELY with the JSON response
    without any preceding text.

Wikipedia titles:
```

First, that prompt was used four separate times across the whole dataset to yield four different result batches each containing a prompt and five descriptions. The prompts were then deduplicated and, for the minority of topics for which multiple distinct prompts remained, the best topic prompt was chosen by asking Claude to decide with this prompt:

```
You are helping to create a high-quality
    dataset for training language models.
    The dataset consists of conversational
    prompts about various topics along
    with multiple labels/descriptions for
    each topic.

For some topics, we have generated
    multiple different prompts. Your task
    is to select or suggest the BEST
    prompt for each topic.

A good prompt should:
1. Be natural and conversational (not
    overly formal)
2. Be clear and unambiguous about what
    topic is being requested
3. Match the style: "Tell me about
    [topic]." or similar casual phrasing
4. Properly handle punctuation,
    capitalization, and formatting for the
    topic name
5. Be consistent with how the topic
    would naturally be discussed

Below is a topic with multiple prompt
    options and its labels. Please respond
    with ONLY the best prompt (or a better
    version if you can improve on the
    options). Do not include any
    explanation or other text.

Original Title: {original_title}
```

```
  Prompt Options:
  {prompt_options}

  Labels:
  {labels}

  Best Prompt:
```

This resulted in a combined raw dataset of 50,005 topics each with a prompt and anywhere from 6–20 descriptions (20 if all four runs produced disjoint sets of descriptions, 6 in a case where the outputs were near-identical). The mean number of descriptions per topic was 16.9.

The vast majority of these topics were interpreted consistently by Claude and resulted in description sets all clearly describing the same topic, but there were some exceptions where Claude misinterpreted the topic in different conflicting ways. For example, "Left Ginza" as an article title refers to one of the two parts of the Ginza Rabba, the scripture of Mandaeism, but in some of its responses Claude hallucinated that Left Ginza was a sub-district of the Ginza district in Tokyo. (Note that if Claude misinterpreted a Wikipedia article title but always in the same consistent way, that's perfectly fine for our purposes, since we only want a diverse dataset of well-known topics. It only degrades the dataset quality if the descriptions are inconsistent.) To weed out such inconsistent records from the dataset, a final filtering step was used with this Claude prompt judging the consistency:

```
You are evaluating the quality of a
    dataset used for training language
    models.

Each dataset entry contains:
- An original Wikipedia article title
- A prompt asking about that topic
- Multiple labels/descriptions that should
    all refer to the same topic

Your task is to evaluate: **How certainly
    do all these labels refer to the same
    unique, real topic?**

Consider:
- Do all labels describe the same
    entity/concept/thing?
- Or do they describe different things
    that happen to have similar names?
- Are the labels internally consistent
    with each other?
- Would someone reading all these labels
    clearly understand what single topic
    is being referenced?

Respond with a JSON object containing:
1. "reasoning": A brief (1-3 sentences)
    explanation of your evaluation
2. "score": A number from 0-10 where:
    - 0 = Labels clearly refer to
```

```
    completely different topics
    (incoherent)
    - 5 = Ambiguous or mixed; some labels
    might refer to different topics
    - 10 = All labels clearly and
    unambiguously refer to the same unique
    topic (highly coherent)

Entry to evaluate:

Original Title: {original_title}

Prompt: {prompt}

Labels:
{labels}

Respond with ONLY valid JSON, no
    additional text:
```

The dataset was filtered to contain only records judged 9 or 10. This resulted in a final validated dataset containing 49,637 topics (99.3% retention), each with a natural "Tell me about [topic]." prompt and 6–20 varied descriptions that unambiguously refer to the same topic. This dataset has been published at https://huggingface.co/datasets/keenanpepper/fifty-thousand-things.

For retrieval evaluation, each topic is represented in the search index by a single document embedding. The document contains the article title followed by a bulleted list of all descriptions for that topic, providing diverse lexical anchors that make retrieval robust to variation in phrasing.

### A.3. SelfIE Prompt Template

We use the following template for all self-interpretation experiments:

User message:

```
What is the meaning of
    "<|reserved_special_token_0|>"?
```

Assistant message (to be continued by autoregressive generation):

```
The meaning of
    "<|reserved_special_token_0|>" is "
```

Note that no system message is used. The Llama models we use handle this well but other models may have a stronger expectation that a system prompt be present, in which case this would have to be modified.

The usage of <|reserved_special_token_0|> as the placeholder was intended to prevent any possible leakage of semantic content from the value of the placeholder token into the description, but since we always inject soft tokens

at the initial embedding layer this is actually not necessary (it would only come into play when injecting at a later layer because then some layers would still see the unpatched, original value of the token embedding for the placeholder token).

The transformed activation $f(h)$ is injected at the placeholder token position at layer 0 (the embedding layer). Generation continues until the model produces a closing quote and end-of-turn token.

### A.4. Training Hyperparameters

*Table 5.* Training hyperparameters for Llama-8B adapter training on a single A100. Epoch count varied by dataset size: 1 epoch for large datasets (Wikipedia, $\sim$840k descriptions) and up to 5 epochs for smaller datasets.

| Hyperparameter | Value |
|---|---|
| Optimizer | AdamW |
| Learning rate | 0.01 |
| Batch size | 256 |
| Epochs | 1–5 (dataset-dependent) |
| Weight decay | 0.01 |
| LR schedule | Cosine decay |
| Warmup steps | 10 |
| Gradient clip norm | 0.5 |
| Initial scale ($\alpha$) | 5.0 |
| Random seed | 42 |

See the table for Llama-8B training hyperparameters. For the Llama-70B training on Goodfire the only differences were decreasing the batch size to 128, increasing the initial scale to 30.0, and running on $3\times$ A100s.

### A.5. Scale Grid and Best-of-N Protocol

All input vectors are normalized to unit L2 norm during training. During generation, we apply scaling factors from the grid $\{0.1, 0.2, 0.3, 0.5, 0.8, 1.3, 2.1, 3.4, 5.5, 8.9, 14.4, 23.3\}$ (approximately geometric with ratio $\phi \approx 1.618$). For adapters that learn their own scale parameter $\alpha$, the net effect is that scales multiply: the external scaling factor modulates the learned scale.

For evaluation, we select the best-performing set of $N = 6$ consecutive scales (for each combination of SelfIE method and dataset) via a calibration run on a small subset, then use those $N$ scales to generate $N$ candidate labels per vector. The baselines (auto-interp labels repeated $N$ times, and the original auto-interp label plus $N - 1$ LLM paraphrases) provide $N$ candidate labels for a fair comparison.

**Two-phase scoring protocol.** For the SAE evaluation metrics in Tables 2 and 3 (generation scoring hit rate and `delphi` detection F1), we use a two-phase procedure that decouples candidate *selection* from final *reporting*:

1. **Selection.** Score all $N$ candidate labels per latent with a first-pass run of the metric (using a fixed seed), and select the candidate with the highest first-pass score.

2. **Rescoring.** Take the single selected label per latent and rescore it under a fresh independent seed. Report the average of these rescored values.

A naive "best-of-$N$" reporting (using the max first-pass score directly) conflates two distinct effects: genuine improvement from having a better label, and upward bias from selecting on the noisier of two random samples of the same underlying score. The latter inflates results even when the $N$ candidates are identical, as is the case for the "Auto-interp Labels $\times 6$" baseline, which would otherwise appear artificially competitive purely due to selection-on-noise. The two-phase protocol eliminates this bias: the rescoring seed is independent of the selection seed, so the reported value is an unbiased estimate of the quality of the selected label.

Selection and rescoring always use the same metric: generation scoring hit rate is selected and reported with hit rate; detection F1 is selected and reported with F1. Coverage values are carried forward unchanged from the original best-of-$N$ runs, as coverage is much less sensitive to selection-on-noise (it depends only on whether any nonzero activation occurs across the generations for the selected label).

For the Wikipedia retrieval columns (R@1, R@100) in Table 2, we use the simpler best-of-$N$ protocol: among the $N$ candidate labels, the best embedding-based retrieval rank is reported. Retrieval ranks are deterministic given a fixed query and embedding index, so there is no scoring noise to select on; the two-phase protocol would degenerate to plain best-of-$N$ here, producing identical numbers.

Figure 4 shows histograms of valid scales per latent on Llama-3.1-8B-Instruct. Trained adapters substantially increase the number of scales producing accurate outputs, partially mitigating the sensitivity issue identified by Kharlapenko et al. (2024).

### A.6. Generation Scoring Protocol

Our implementation of generation scoring uses the following prompt to generate the synthetic contexts to be evaluated for SAE latent activation. Unlike the SelfIE prompt, this is an ordinary (hard) prompt with no soft tokens.

System message:

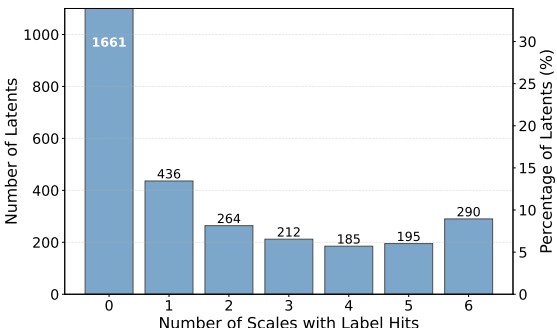

*(a)* Scale sensitivity histogram for untrained SelfIE

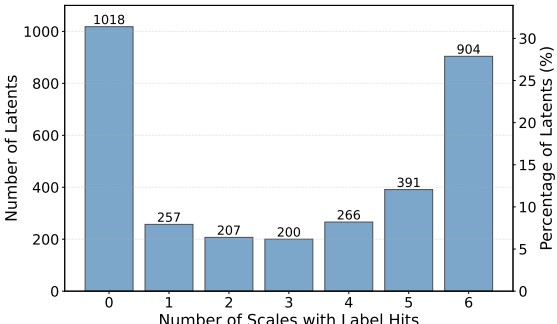

*(b)* Scale sensitivity histogram for trained SelfIE (scalar affine + low rank, r=64, Llama Scope dataset)

*Figure 4.* Histograms showing the distribution of the number of scales (out of 6 scales attempted) at which each method produced accurate labels (where "accurate" is defined as eliciting at least one nonzero activation in 10 trials of generation scoring). The trained adapter is less sensitive to scale, with more latents receiving accurate labels at all 6 scales.

```
You are a helpful AI assistant who
    generates EXTREMELY SHORT example
    conversations. The conversations are
    between a user and an assistant, and
    have the following format:
[USER] I'm a user.
[ASSISTANT] I'm the assistant.
```

User message:

```
Produce a VERY SHORT conversation which
    exhibits '{{LABEL}}'
Do not include any other text in your
    response. Start immediately with the
    conversation.
```

The same experimental model then auto-regressively generates a quoted conversation using the [USER] and [ASSISTANT] tags. We use nucleus sampling (Holtzman et al., 2019) at temperature 0.7 with $p = 0.9$. After generation, the quoted conversation is algorithmically converted to a real conversation in the Llama chat template format. If any quoted conversations did not adhere to the expected format, they were interpreted by default as a single verbatim assis-

tant message, but such parse errors occurred on only 0.04% of all generations. Then a single forward pass is performed to get the SAE activations. The very first set of activations (for the `<|begin_of_text|>` token) is discarded, since the Llama Scope SAE was not trained on this token and there are many spurious activations.

The actual magnitudes of the nonzero activations can vary widely for different SAE latents, and also the latents have different activation patterns where some tend to activate on single tokens where others remain active for many tokens. Therefore our chosen metrics do not distinguish between different nonzero activation values, or even how many tokens within a generation had nonzero activations, but instead depend entirely on whether each overall generation is a 'hit' (some nonzero activation other than the `<|begin_of_text|>`) or a 'miss' (no nonzero activations except possibly `<|begin_of_text|>`). The two metrics we use are:

- Hit rate: Fraction of generations, for a single label, that had a 'hit'

- Coverage: Simply 1 if *any* generation for a label was a 'hit' and 0 otherwise

In Tables 2 and 3, the hit rate is reported under the two-phase selection-and-rescore protocol described in Appendix A.5, while the coverage is reported as the mean (over all SAE latents evaluated) of the per-label best-of-$N$ coverage.

### A.7. Detection Scoring Protocol

We use the `delphi` library (Paulo et al., 2024) for detection scoring. For the Llama-3.1-8B Llama Scope SAE evaluation in Table 2, we use Llama-3.1-8B-Instruct itself as the judge model and the pre-compiled activation contexts available from Neuronpedia. For the Llama-3.3-70B Goodfire SAE evaluation in Table 3, we use Llama-3.3-70B-Instruct as the judge model and the corresponding Goodfire 70B activation caches available from Neuronpedia (no public detection-scoring activation contexts are available for the Goodfire 8B SAE, which is why detection F1 is not reported for that dataset in Table 2). Given a label, a classifier based on prompting the judge model with the label determines whether text snippets would activate the latent. The F1 score measures agreement with actual activations. This differs from generation scoring in that it uses a pre-compiled dataset of contexts, while generation scoring creates its own contexts to evaluate.

Detection F1 in Tables 2 and 3 is reported under the same two-phase select-then-rescore protocol as the generation scoring hit rate (Appendix A.5), with F1 used as the metric for both phases.

## A.8. Computational Resources

All experiments were conducted on NVIDIA A100 80GB GPUs.

**Training.** Table 6 shows per-run training times. We estimate total training compute at approximately 50–80 GPU-hours across all adapter architectures and datasets explored in this work.

*Table 6.* Training time per adapter (A100 80GB).

| Model size | Time | GPUs |
|---|---|---|
| 7B–14B | 4–6 h | 1 |
| 32B | 6–8 h | 2 |
| 70B–72B | 9–15 h | 3 |

**Evaluation.** Inference compute was measured empirically on Llama-3.1-8B-Instruct (throughput of 1,521 tokens/second at batch size 512), then extrapolated to larger models assuming throughput roughly inversely proportional to parameter count. Total evaluation compute across all experiments was approximately 136 GPU-hours: generation scoring on SAE latents (∼72 GPU-hours, dominated by the 70B evaluation at 59 GPU-hours), topic retrieval evaluations across the Qwen scaling series (∼53 GPU-hours), and bridge entity detection (∼12 GPU-hours for 3.2M SelfIE generations across 500 prompts × 32 layers × 20 token positions).

Total compute across all experiments was approximately 180–220 GPU-hours (training plus evaluation).

**Dataset generation.** The Wikipedia topic descriptions (∼840k labels across 50k topics) were generated using Claude Sonnet, requiring approximately $300 in API costs.

## B. Training Dynamics

Figure 5 shows training dynamics for different adapter architectures on Llama Scope SAE data. Full-rank adapters begin overfitting partway through the first epoch (although this overfitting is not yet visible since both validation samples and train samples are never-before-seen by the model, so their average losses are equal). After the first epoch this gap is clearly visible, but even at the end of the first epoch it manifests in the full-rank adapter having higher loss than the SA+LR adapter (validation loss 1.691 vs 1.661, with the train loss closely tracking it). The SA+LR adapter also has a train-loss gap that increases in successive epochs, but in this case the overfitting is not catastrophic and at the end of the *third* epoch this adapter has the lowest validation loss ever seen in the training of any run in Table 1.

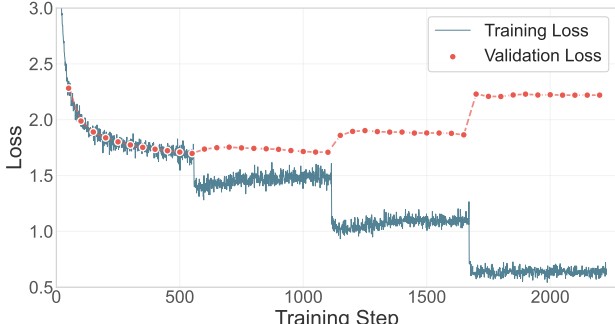

*(a)* Training Progress (Full-Rank)

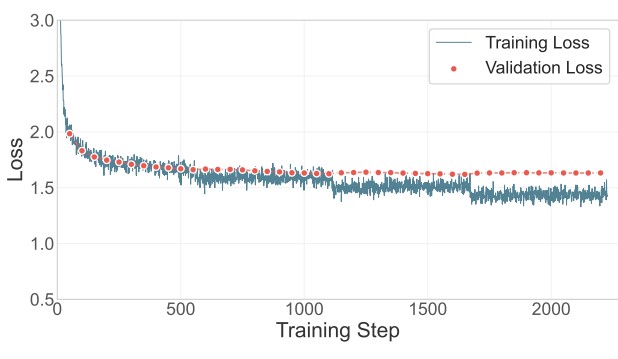

*(b)* Training Progress (SA+LR)

*Figure 5.* Validation loss curves during training on Llama Scope SAE features. Full-rank adapters (a) achieve lower training loss but higher validation loss than scalar affine + low-rank adapters (b), demonstrating overfitting. The train-loss gap only appears after the first epoch, but actually the full-rank adapter is already underperforming at the end of the first epoch (validation loss 1.691 rather than 1.661).

### B.1. Effect of Data Shuffling

We compared training with data reshuffled each epoch versus maintaining the same order throughout. Figure 6 shows that fixed ordering produces smoother loss curves while reshuffling creates visible stair-step patterns at epoch boundaries. However, both conditions exhibit similar final overfitting behavior for full-rank adapters, confirming that overfitting stems from excessive model capacity rather than memorization of presentation order.

## C. Training Data Requirements

To characterize how much training data is needed for effective self-interpretation, we trained scalar affine adapters on varying fractions of the Goodfire-8B SAE dataset and evaluated on both in-distribution (Goodfire) and cross-dataset (Llama Scope) generation scoring.

Figure 7 shows that performance improves steadily with

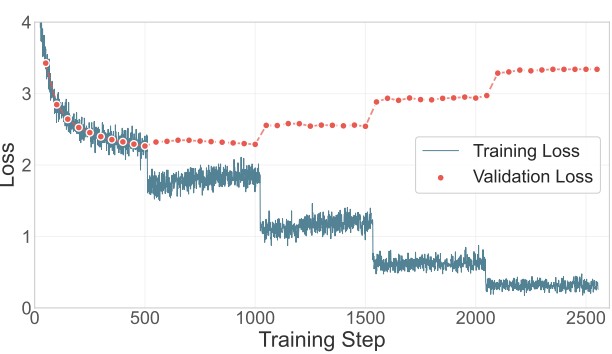

*(a)* Training Progress (Full-Rank, Shuffled)

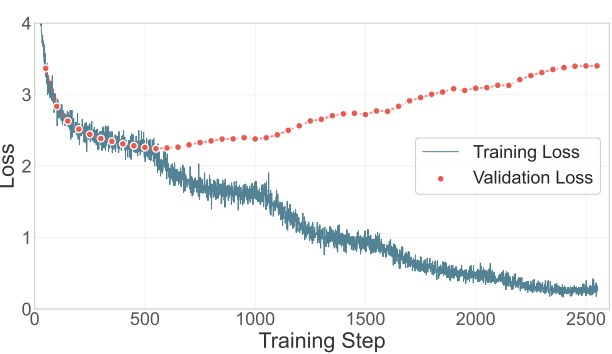

*(b)* Training Progress (Full-Rank, Unshuffled)

*Figure 6.* Validation loss curves for full-rank adapters with different shuffling strategies. Reshuffling each epoch (a) produces stair-step patterns as the model encounters some fraction of very-recently-seen samples at the beginning of each epoch; using the same order each epoch (b) produces smooth curves. Both overfit similarly, confirming that overfitting is due to model capacity rather than ordering effects.

training data but exhibits diminishing returns. Notably, the cross-dataset generalization curve tracks the in-distribution curve closely, indicating that additional training data improves genuine self-interpretation capability rather than overfitting to dataset-specific patterns.

# D. Contrastive Topic Vectors Architecture Comparison

On Wikipedia contrastive vectors, full-rank adapters achieve the best performance (Table 7). This contrasts sharply with SAE features, where full-rank overfits catastrophically. Why does full-rank succeed on Wikipedia but overfit on SAEs? We hypothesized two possible explanations: (1) Wikipedia vectors have lower intrinsic dimensionality, providing implicit regularization, or (2) Wikipedia has more labels per vector ($\sim$17 vs 1), providing more supervision.

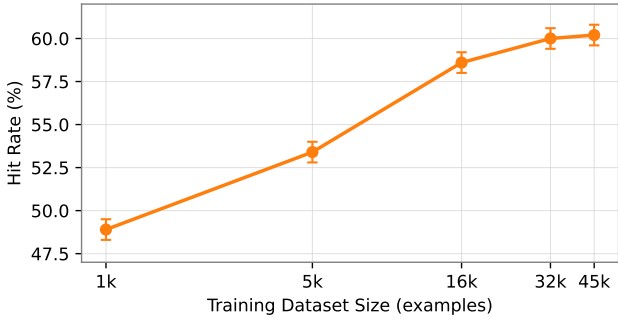

*(a)* Goodfire (In-Distribution)

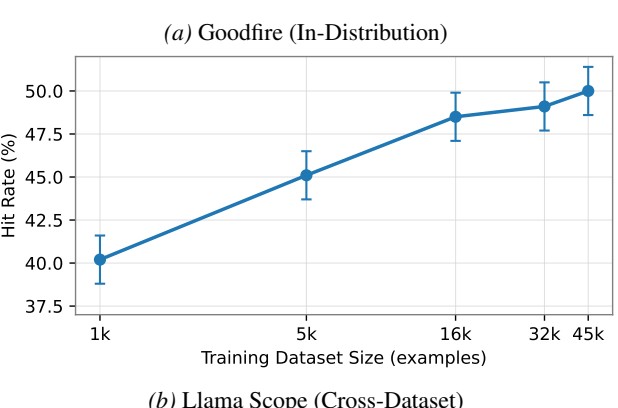

*(b)* Llama Scope (Cross-Dataset)

*Figure 7.* Performance of scalar affine adapters trained on varying fractions of the Goodfire-8B SAE training set. In-distribution evaluation (a) and cross-dataset evaluation (b) both show performance improving with more training data, with diminishing returns beyond 20k–30k labeled vectors. Cross-dataset generalization follows a similar curve, suggesting that the adapter learns genuinely transferable structure rather than dataset-specific patterns.

## D.1. PCA Analysis

Figure 8 shows PCA analysis of both datasets. Wikipedia vectors are effectively low-dimensional, while SAE features utilize the full embedding space.

## D.2. Controlled Experiments

To distinguish intrinsic dimensionality from label count, we conducted two controlled experiments:

**Wikipedia with 1 label per vector.** We trained full-rank adapters on Wikipedia using only a single label per vector (matching SAE's label count). Validation loss continued decreasing throughout training with no sign of overfitting, ruling out label count as the explanation.

**SAE with 6 labels per vector.** We augmented Goodfire SAE latents with 5 LLM-generated paraphrases per latent, yielding 6 total labels per vector. Full-rank adapters still overfitted, with validation loss increasing partway through epoch 1.

These experiments appear to rule out label count as the rea-

*Table 7.* Architecture comparison on Wikipedia contrastive vectors (Llama-3.1-8B). Unlike SAE features, full-rank adapters achieve the best performance without overfitting.

| ARCHITECTURE | PARAMS | VAL LOSS | Δ |
|---|---|---|---|
| IDENTITY | 0 | 3.858 | — |
| SCALE-ONLY | 1 | 3.523 | -0.335 |
| SCALAR AFFINE | 4097 | 1.366 | -2.492 |
| SA + LR (R=4) | 37K | 1.293 | -2.565 |
| SA + LR (R=16) | 135K | 1.247 | -2.611 |
| SA + LR (R=64) | 528K | 1.205 | -2.653 |
| SA + LR (R=256) | 2.1M | 1.195 | -2.663 |
| LR ONLY (R=4) | 37K | 1.811 | -2.047 |
| LR ONLY (R=16) | 135K | 1.415 | -2.443 |
| LR ONLY (R=64) | 528K | 1.247 | -2.611 |
| LR ONLY (R=256) | 2.1M | 1.217 | -2.641 |
| FULL-RANK AFFINE | 16.8M | **1.160** | **-2.698** |

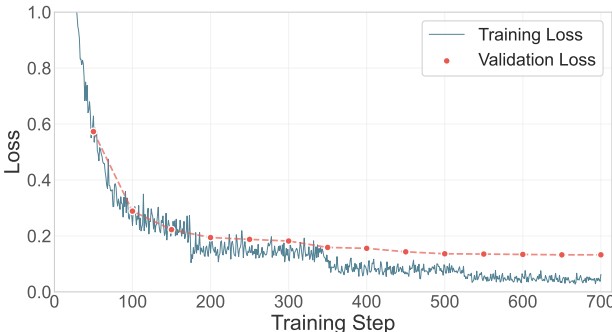

*Figure 9.* Loss curves for training a full-rank adapter on Wikipedia topic contrastive vectors with only a single text label per vector (the article title). A train-val gap does appear; nevertheless the validation loss continues to decrease over the whole training run.

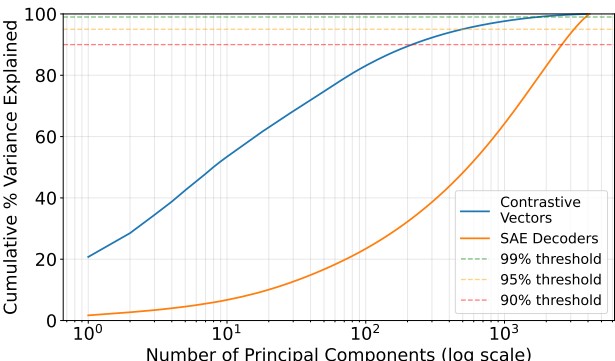

*Figure 8.* Cumulative variance explained by principal components. Wikipedia contrastive vectors (blue) concentrate >90% of variance in approximately 200 dimensions, while SAE decoder vectors (orange) span nearly the full 4096-dimensional space.

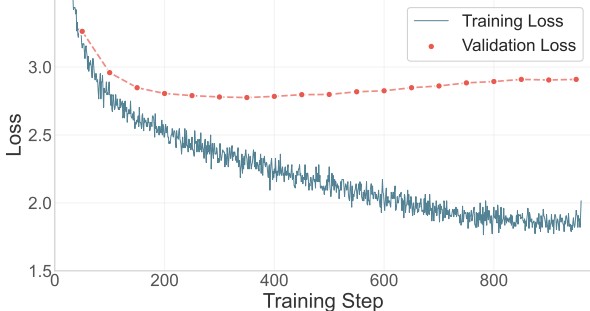

*Figure 10.* Loss curves for training a full-rank adapter on Goodfire SAE latents with 6 text labels per vector (the original Goodfire auto-interp label + 5 LLM paraphrases). The validation loss reaches a minimum about 36% of the way through the first epoch and then rises.

son why full-rank adapters overfit on SAE latents but not on contrastive vectors, isolating intrinsic dimensionality as the operative factor. We conjecture that when the training dataset consists of some tens of thousands of vectors that span a large fraction of the dimensionality of the space, it's possible for a full-rank adapter to learn what's essentially a lookup table, promoting memorization, while if a similar-sized dataset has an intrinsic dimensionality in the low hundreds, an affine map is limited to learning functions on this much smaller subspace. In this case the *effective* parameter count is reduced from 16M to <1M promoting generalizable interpretation of the topic vectors.

# E. Cross-Layer Generalization

We trained scalar affine adapters on Llama Scope 32k SAEs either independently at each layer or on vector-label pairs pooled across all 32 layers. Table 8 and Figure 11 show that a single cross-layer adapter matches and even slightly

outperforms 32 layer-specific adapters. This was surprising to us and indicates that the geometric-semantic correspondence of the residual stream is preserved remarkably well over different layers: a SelfIE adapter doesn't need to know which layer an activation comes from in order to interpret it successfully. Another explanation consistent with this finding is that the bias vector mostly acts as a layer-agnostic soft token that steers the language model generation in directions that are generally useful for this interpretation task, and not as a layer-specific distributional shift.

The Qwen-2.5 scaling experiment (Appendix F) provides an even more dramatic demonstration of this cross-layer generalization phenomenon. In that experiment, full-rank affine adapters were trained on contrastive activation vectors pooled across all layers in the middle half of models ranging from 7B to 72B parameters. These adapters successfully interpret activations from any layer in their training set without knowing which layer the activation came from, definitively confirming that this remarkable generalization

*Table 8.* Comparison of layer-specific vs cross-layer training, evaluated on Llama Scope SAE features. The cross-layer adapter actually outperforms the 32 layer-specific adapters on the hit rate metric, to a small but statistically significant extent (paired t-test, $p = 0.0002$). Because this experiment evaluates 32 layers × multiple adapter configurations, we use a cheaper evaluation protocol than Table 2: 4 scales × 1 trial each, combined disjunctively (any-scale-hits = hit). This is equivalent to plain best-of-$N$ reporting and is *not* the two-phase select-then-rescore protocol used elsewhere in the paper, so hit rates here are inflated by selection-on-noise and should not be directly compared to Table 2 or 11; the comparison between rows of this table is internally consistent because all rows use the same protocol.

| Training data | Hit rate | Coverage |
|---|---|---|
| Untrained SelfIE | $41.0\%_{\pm 0.2}$ | $70.2\%_{\pm 0.3}$ |
| Layer-specific (eval on same layer) | $64.7\%_{\pm 0.2}$ | $89.2\%_{\pm 0.2}$ |
| Cross-layer (layers 0–31 combined) | $65.7\%_{\pm 0.2}$ | $89.4\%_{\pm 0.2}$ |

extends beyond the minimal scalar affine architecture to adapters with millions of parameters.

## F. Scaling Experiment Details

This appendix provides additional details and metrics for the Qwen-2.5 scaling experiment (Section 3.4).

### F.1. Experimental Setup

For each Qwen-2.5 model (7B, 14B, 32B, 72B), we computed contrastive activation vectors for the Wikipedia topics dataset at each layer in the middle half of the model. Vectors from all these layers were pooled together and shuffled, then used to train a single full-rank affine adapter. The adapter receives only the activation vector as input, with no information about which layer it came from.

The "Taboo" baseline uses the following prompt:

```
Describe {topic_phrase} without using the
    word "{original_title}", any part of
    it, or obvious synonyms. Be specific
    enough that someone could guess what
    you're describing.
```

*e.g.*

```
Describe bananas without using the word
    "Banana", any part of it, or obvious
    synonyms. Be specific enough that
    someone could guess what you're
    describing.
```

Both SelfIE generations and Taboo descriptions are scored using GTE-large embeddings with recall@$k$ retrieval against all ~50k topics.

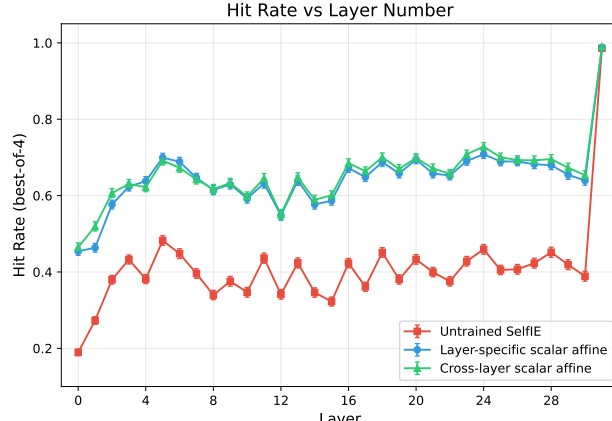

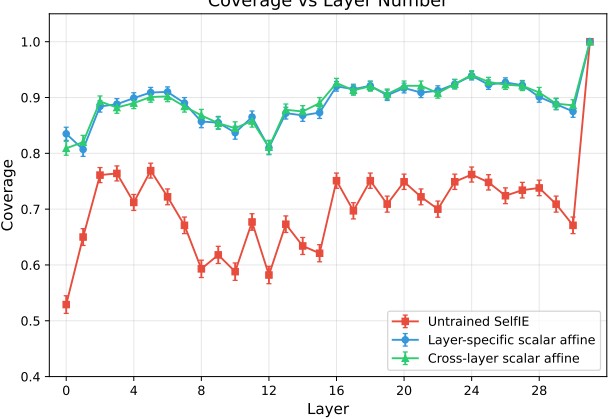

*Figure 11.* Comparison of layer-specific vs cross-layer scalar affine adapters, performance by layer.

### F.2. Additional Metrics

Beyond the recall@100 metric shown in Figure 2, we examine additional aspects of retrieval performance. Figures 12 and 13 show recall@1 and Mean Reciprocal Rank (MRR) for the same adapters, also averaged over all layers in the middle half of the model, on which each adapter was trained.

Another way to incorporate the results from the Taboo comparison is to **filter** the topics analyzed to only those where the Qwen model succeeds at describing the topic well enough that it's the top search result (recall@1). In order to compare the four SelfIE adapters on the same dataset, we therefore take the **intersection** of the Taboo recall@1 sets for each of the four models. The results are shown in Figure 14.

When evaluating on the single best-performing layer per model rather than averaging across the middle half (Figure 15), the 32B model breaks the otherwise monotonic scaling trend, achieving lower recall than 14B (32.3% vs 39.9%). Two architectural observations may explain this: First, Qwen2.5-14B and Qwen2.5-32B share identical residual stream dimensions (hidden_size=5120), meaning the

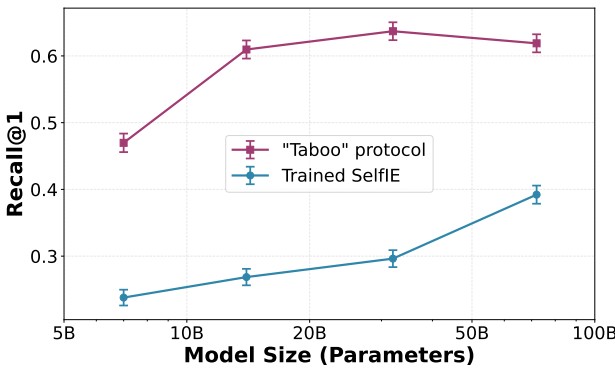

*Figure 12.* Recall@1 comparison between SelfIE and Taboo baseline across model scales, averaged over all layers in the middle half of each model. The same scaling trend observed in recall@100 holds for this stricter metric: SelfIE performance increases more rapidly than the Taboo baseline, narrowing the gap at larger model scales.

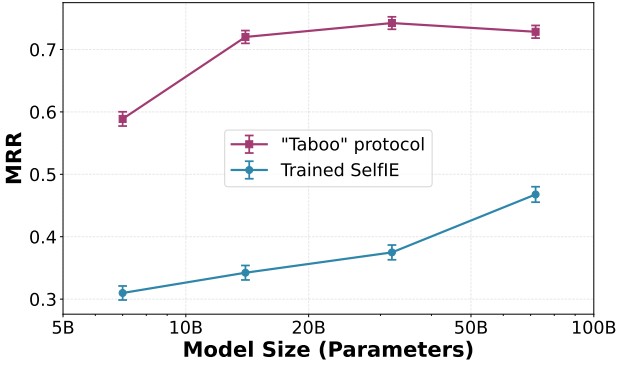

*Figure 13.* Mean Reciprocal Rank (MRR) comparison between SelfIE and Taboo baseline across model scales, averaged over all layers in the middle half of each model. SelfIE approaches the Taboo upper bound as model scale increases.

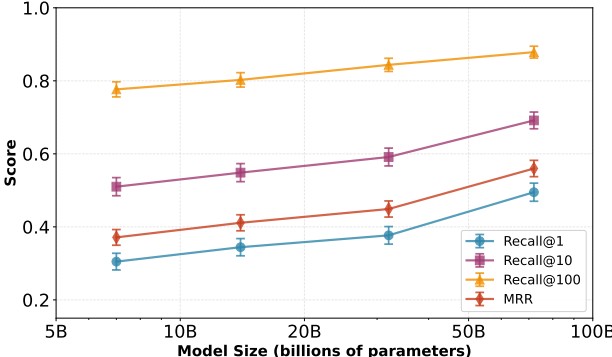

*Figure 14.* SelfIE results across scales, but limited to the subset of 1559 topics "known" by all four Qwen models in that they all achieve recall@1 on the Taboo baseline.

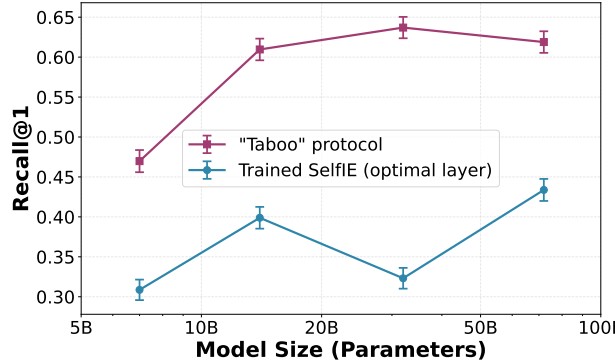

*Figure 15.* Recall@1 performance evaluated on the single best-performing layer for each model scale. Unlike the averaged metrics, this view reveals a non-monotonic scaling trend where the 32B model underperforms the 14B model, likely due to architectural differences in residual stream dimensions and optimal layer depth.

full-rank adapter has identical capacity for both models despite the 2.3× parameter difference. Second, the optimal layer for 32B occurs at 38% relative depth (layer 24 of 64), substantially shallower than the other models (63–73% relative depth). This suggests that semantic representations suitable for self-interpretation concentrate earlier in 32B's forward pass, possibly because the additional layers (64 vs 48) primarily perform output refinement rather than semantic enrichment. When averaging across all middle-half layers, the scaling trend is monotonic, indicating that this anomaly is specific to single-layer evaluation.

# G. Generalization to Other Model Families

To verify that our SAE findings generalize beyond the Llama model family, we trained adapters on Gemma-2-9B-IT using Gemma Scope SAEs (Lieberum et al., 2024). Tables 9 and 10 show architecture comparisons for 16k and 131k

width SAEs respectively.

The results broadly replicate our Llama findings. On the smaller 16k SAE, scalar affine achieves the best validation loss, with low-rank additions providing no benefit, which makes sense because smaller datasets are easier to overfit. On the larger 131k SAE, the pattern matches Llama more closely: SA+LR (r=64) achieves the best performance, with the identity-preserving structure providing consistent gains over low-rank only variants. The bias vector remains critical in both cases, with scalar affine improving by over 3.4 loss units.

## G.1. Gemma-2-9B-IT Evaluation

We also evaluated trained adapters on generation scoring and cross-dataset transfer.

**Generation scoring on Gemma Scope SAEs.** Table 11 shows generation scoring on held-out Gemma Scope la-

*Table 9.* Architecture comparison on Gemma Scope 16k SAE (layer 20). Parameter counts for Gemma-2-9B-IT ($d_{\text{model}} = 3584$). Scalar affine achieves the best validation loss.

| ARCHITECTURE | PARAMS | VAL LOSS | $\Delta$ |
|---|---|---|---|
| IDENTITY | 0 | 5.382 | — |
| SCALE-ONLY | 1 | 5.377 | -0.005 |
| SCALAR AFFINE | 3585 | **1.960** | **-3.422** |
| SA + LR (R=4) | 32K | 1.976 | -3.405 |
| SA + LR (R=16) | 118K | 2.024 | -3.358 |
| SA + LR (R=64) | 462K | 2.075 | -3.307 |
| SA + LR (R=256) | 1.8M | 2.156 | -3.226 |
| LR ONLY (R=4) | 32K | 2.221 | -3.161 |
| LR ONLY (R=16) | 118K | 2.157 | -3.225 |
| LR ONLY (R=64) | 462K | 2.144 | -3.237 |
| LR ONLY (R=256) | 1.8M | 2.211 | -3.171 |
| FULL-RANK AFFINE | 12.8M | 2.327 | -3.055 |

*Table 10.* Architecture comparison on Gemma Scope 131k SAE (layer 20). SA+LR (r=64) achieves the best validation loss, matching the pattern observed on Llama Scope SAEs.

| ARCHITECTURE | PARAMS | VAL LOSS | $\Delta$ |
|---|---|---|---|
| IDENTITY | 0 | 5.478 | — |
| SCALE-ONLY | 1 | 5.454 | -0.024 |
| SCALAR AFFINE | 3585 | 2.008 | -3.471 |
| SA + LR (R=4) | 32K | 1.932 | -3.547 |
| SA + LR (R=16) | 118K | 1.890 | -3.589 |
| SA + LR (R=64) | 462K | **1.879** | **-3.600** |
| SA + LR (R=256) | 1.8M | 1.947 | -3.531 |
| LR ONLY (R=4) | 32K | 2.151 | -3.328 |
| LR ONLY (R=16) | 118K | 1.979 | -3.500 |
| LR ONLY (R=64) | 462K | 1.896 | -3.583 |
| LR ONLY (R=256) | 1.8M | 1.968 | -3.511 |
| FULL-RANK AFFINE | 12.8M | 1.986 | -3.493 |

*Table 11.* Generation scoring on Gemma Scope SAE latents (Gemma-2-9B-IT, layer 20, 131k SAE; $n = 1000$ held-out latents). Hit rates are reported under the same two-phase select-then-rescore protocol as Table 2 (see Appendix A.5), with Phase-2 $N = 10$. GemmaScope-trained adapters improve over untrained SelfIE; Wikipedia-trained adapters show poor transfer.

| METHOD | HIT RATE |
|---|---|
| ORIGINAL + 5 PARAPHRASES | $39.1_{\pm 1.3}$ |
| AUTO-INTERP LABELS ×6 | $32.6_{\pm 1.3}$ |
| UNTRAINED SELFIE | $28.5_{\pm 1.3}$ |
| *GemmaScope-trained:* | |
| SCALAR AFFINE | $\mathbf{38.9}_{\pm 1.4}$ |
| FULL-RANK | $37.0_{\pm 1.3}$ |
| SA+LR (R=64) | $32.8_{\pm 1.3}$ |
| *Wikipedia-trained:* | |
| SA+LR (R=64) | $26.1_{\pm 1.2}$ |
| SCALAR AFFINE | $17.1_{\pm 1.0}$ |
| FULL-RANK | $16.5_{\pm 1.0}$ |

ization.

*Table 12.* Wikipedia topic retrieval on Gemma-2-9B-IT (best-of-6 protocol). Wikipedia-trained adapters achieve strong retrieval; GemmaScope-trained adapters show minimal transfer.

| TRAINING DATA | ARCHITECTURE | R@1 | R@100 |
|---|---|---|---|
| WIKIPEDIA | FULL-RANK | **81.3%** | **98.5%** |
| WIKIPEDIA | SA+LR (R=64) | 74.1% | 98.1% |
| WIKIPEDIA | SCALAR AFFINE | 46.9% | 93.3% |
| GEMMASCOPE | SCALAR AFFINE | 3.2% | 46.5% |
| GEMMASCOPE | FULL-RANK | 0.4% | 17.7% |
| GEMMASCOPE | SA+LR (R=64) | 0.3% | 14.7% |

These results confirm that our core findings replicate across model families: trained adapters improve self-interpretation, training data matching matters for generalization, and full-rank succeeds on Wikipedia but not SAEs.

### G.2. Qwen

Unlike Llama and Gemma, no high-quality public SAEs exist for Qwen at time of writing. However, this limitation demonstrates a strength of our approach: training on Wikipedia contrastive vectors requires only the model itself, with no external interpretability artifacts. We trained adapters on Qwen-2.5-7B-Instruct using our Wikipedia topics dataset and evaluated via embedding-based retrieval.

Table 13 shows the same patterns: full-rank adapters achieve the best retrieval performance (52.6% R@1), with the architecture hierarchy matching Llama and Gemma. This confirms that trained self-interpretation generalizes to any open-weights model without requiring pre-existing SAEs. The Wikipedia contrastive vectors dataset serves as a universal training signal applicable across model families.

tents. Trained adapters improve over untrained SelfIE (38.9% vs 28.5%), though the gap is smaller than on Llama. GemmaScope-trained adapters outperform Wikipedia-trained adapters on this in-distribution task, consistent with our finding that training data matching matters more than architecture.

**Scale sensitivity.** Figure 16 shows histograms of valid scales per latent. As with Llama, trained adapters substantially increase the number of scales producing accurate outputs.

**Wikipedia topic retrieval.** Table 12 shows embedding-based retrieval on Wikipedia topics. Gemma full-rank adapters achieve the best performance on Wikipedia topics (81.3% R@1), matching the Llama pattern for this dataset. As expected, GemmaScope-trained adapters show minimal transfer to Wikipedia topics (<3.5% R@1), confirming that training data matching is essential for cross-dataset general-

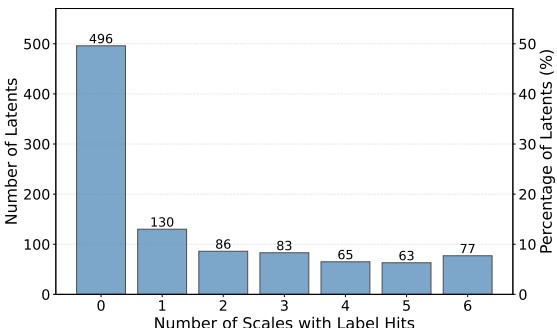

*(a)* Scale sensitivity histogram for untrained SelfIE

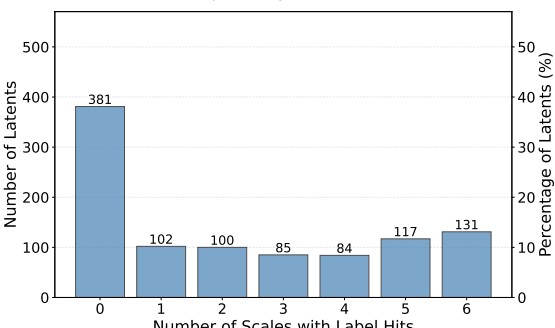

*(b)* Scale sensitivity histogram for trained SelfIE (scalar affine, Gemma Scope dataset)

*Figure 16.* Histograms showing the distribution of the number of scales (out of 6 scales attempted) at which each method produced accurate labels (where "accurate" is defined as eliciting at least one nonzero activation in 10 trials of generation scoring). The trained adapter is less sensitive to scale, with more latents receiving accurate labels at all 6 scales.

# H. Bridge Entity Detection Details

## H.1. Immediate Answering Protocol

To verify that the model answers two-hop questions without verbalized chain-of-thought, we use the following prompt template:

```
Complete the following statement with only
   the name of a {category}. If you don't
   know, make your best guess. {prompt}
```

We use one-shot prompting where the assistant's response is pre-filled with a single example answer:

```
User: Complete the following statement
   with only the name of a city. If you
   don't know, make your best guess. The
   capital of the country of origin of
   Tom Clancy's Rainbow Six Siege is
Assistant: Ottawa
```

This one-shot example demonstrates the expected format (a single word or short phrase) and primes the model to respond immediately rather than reasoning step-by-step.

*Table 13.* Wikipedia topic retrieval on Qwen-2.5-7B-Instruct (best-of-6 protocol). Trained adapters achieve strong retrieval despite no SAEs being available for this model family.

| ARCHITECTURE | R@1 | R@100 |
|---|---|---|
| FULL-RANK | **52.6%** | **95.2%** |
| SA+LR (R=64) | 51.3% | 93.4% |
| SCALAR AFFINE | 8.3% | 59.8% |
| UNTRAINED SELFIE | 0.02% | 3.1% |

## H.2. Training Source Comparison

The main text reports bridge entity detection using a Wikipedia-trained adapter. Here we compare against an adapter trained on the Llama Scope SAE to verify the effect generalizes across training sources.

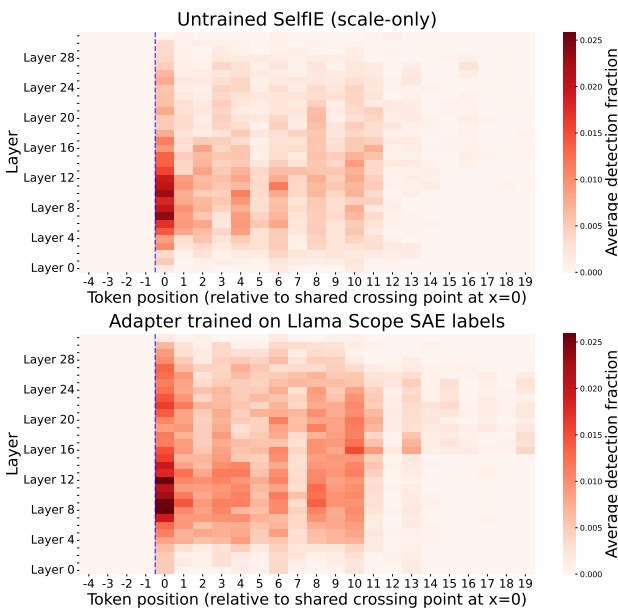

*Figure 17.* Bridge entity detection using SAE-trained scalar affine adapter. Compare to Figure 3 (Wikipedia-trained). Detection rates are lower but still significantly exceed untrained SelfIE.

Training the adapter on Llama Scope SAE labels improves the overall fraction of bridge entities ever detected from $56.4\%_{\pm 2.2}$ to $66.4\%_{\pm 2.1}$, and when only considering the 242 questions where both methods succeeded at detecting the bridge entity, the SAE-trained adapter improves the overall fraction of generations including a match from $0.43\%_{\pm 0.05}$ to $0.71\%_{\pm 0.08}$, an increase of $1.7\times_{\pm 0.1}$.

From inspecting some of the generations from the SAE-trained SelfIE adapter, it's clear that a common failure mode here is for many of the generations to describe or name the current input token rather than higher-level semantic features, even for activations from relatively later layers. This makes sense when we consider that many SAE latents actually describe token-identity-level features and this

*Table 14.* Bridge entity detection rates by training source. Both trained adapters substantially outperform untrained SelfIE; Wikipedia-trained adapters show stronger transfer, possibly because the bridge entity extraction task is closer to being in-distribution for the semantic topic labeling task than it is to the arbitrary SAE latent labeling task.

| Adapter | Prompts detected | Detection rate |
|---|---|---|
| Wikipedia-trained (SA) | 455/500 | $91.0\%_{\pm 1.3}$ |
| SAE-trained (SA) | 332/500 | $66.4\%_{\pm 2.1}$ |
| Untrained SelfIE | 282/500 | $56.4\%_{\pm 2.2}$ |

*Table 15.* Format and accuracy by scale for the ALL-CAPS trained adapter (1000 latents evaluated at each scale). At low scales, outputs are nearly 100% ALL-CAPS; at high scales, nearly 0%. Hit rates (generation scoring) peak at intermediate scales, suggesting a tradeoff between format fidelity and semantic accuracy.

| SCALE | % ALL-CAPS | HIT RATE |
|---|---|---|
| 0.5 | 100.0% | 0.126 |
| 0.8 | 99.9% | 0.283 |
| 1.3 | 99.4% | 0.407 |
| 2.1 | 80.1% | 0.428 |
| 3.4 | 16.8% | 0.385 |
| 5.5 | 1.4% | 0.246 |

adapter was trained to name those, whereas in the case of the Wikipedia topic vectors the adapter was specifically trained to ignore token-level information and focus on extracting the topic-level semantic content.

# I. ALL-CAPS Training Experiments

To understand what the adapter's bias vector encodes, we conducted a controlled experiment: we trained a scalar affine adapter on Goodfire SAE labels that were transformed to ALL-CAPS format. If the bias vector captures format/style information while semantic content comes from the activation vector, we would expect (1) generated labels to be ALL-CAPS when the bias dominates (low scales) and (2) the semantic content to remain accurate regardless of format.

## I.1. Experimental Setup

We took the Goodfire auto-interpretability labels for all available latents and converted them to uppercase. We then trained a scalar affine adapter on these ALL-CAPS labels using the same protocol as our main experiments. At evaluation time, we generated labels at six scales from the standard grid ($\{0.5, 0.8, 1.3, 2.1, 3.4, 5.5\}$) and classified each generation as ALL-CAPS or not based on whether all alphabetic characters were uppercase.

## I.2. Results

Table 15 shows a striking scale-dependent transition in output format. At scale 0.5, 100% of generated labels are ALL-CAPS. This drops to 80% at scale 2.1, 17% at scale 3.4, and just 1.4% at scale 5.5. The transition is remarkably clean: 63% of latents follow the sequence AAAANN (ALL-CAPS at scales 0.5–2.1, not ALL-CAPS at scales 3.4–5.5), with another 19% following AAANNN and 16% following AAAAAN.

Importantly, semantic accuracy (measured by generation scoring hit rate) remains reasonable across the scale range,

peaking at intermediate scales where neither bias nor input vector completely dominates. Compared head-to-head against the standard Goodfire-trained adapter (toasty-mountain) under matched conditions on 1000 held-out latents, the ALL-CAPS adapter achieves $60.7\%_{\pm 1.3}$ Phase-2 hit rate versus $61.8\%_{\pm 1.3}$ for the standard adapter—nearly identical performance despite the unusual output format (both numbers use the same two-phase protocol as Table 2). Filtering the ALL-CAPS adapter's Phase-2 evaluations to count only generations that are themselves in ALL-CAPS format yields $51.1\%_{\pm 1.3}$, a stricter test that requires both format fidelity and semantic accuracy on the same generation.

The two-phase protocol also offers a direct measurement of how often the trained format is preferred by the selection step: 58.2% of the Phase-1-selected labels for the ALL-CAPS adapter are in ALL-CAPS format, compared to only 0.1% for the standard adapter. (Every one of the 1000 latents had at least one ALL-CAPS candidate among its 6 scale variants, so the ALL-CAPS-filtered evaluation drops no latents.) Together these numbers confirm that the ALL-CAPS format itself does not substantially harm semantic accuracy, and that the trained format is the one Phase-1 selection consistently prefers for the ALL-CAPS adapter.

## I.3. Qualitative Example

Table 16 shows generated labels for Goodfire latent #41101—the same latent featured in Table 4. The ALL-CAPS trained adapter produces semantically accurate descriptions ("event handling," "callbacks in programming") that match the standard adapter's output, but in ALL-CAPS format when the scale is low enough for the bias to dominate.

## I.4. Interpretation

These results support a clean decomposition of the adapter's function:

- **The bias vector encodes format and prior.** When

*Table 16.* Greedy-decoded labels for Goodfire latent #41101 (the same latent shown in Table 4) using the ALL-CAPS trained adapter. The ALL-CAPS adapter produces semantically similar descriptions to the standard adapter ("event handling," "callbacks"; compare Table 4) but in ALL-CAPS format at low scales. At high scales (5.5), the semantic content degrades as the input vector overwhelms the bias.

| Scale | Label |
|---|---|
| 0.5 | "THE ASSISTANT IS EXPLAINING HOW TO USE A NEW CONCEPT OR FEATURE" |
| 0.8 | "THE ASSISTANT IS EXPLAINING HOW TO HANDLE OR PROCESS INPUT" |
| 1.3 | "EVENT HANDLING AND CALLBACKS IN PROGRAMMING" |
| 2.1 | "EVENT HANDLING AND CALLBACKS IN PROGRAMMING" |
| 3.4 | "A FUNCTION THAT WILL BE CALLED WHEN SOMETHING HAPPENS... |
| 5.5 | "on" or "at" in English. It's a preposition... |

the input vector is scaled down, the bias dominates, and outputs reflect the training distribution's format (ALL-CAPS) and generic content patterns.

- **The input vector contributes semantic specificity.** The particular latent being interpreted determines the semantic content of the output, regardless of format. The same latent produces descriptions about "event handling" and "callbacks" whether trained on ALL-CAPS or standard labels.

- **Scale controls the interpolation.** Low scales produce outputs dominated by the bias (high format fidelity, generic content); high scales produce outputs dominated by the input vector (format breakdown, but potentially more specific content).

This decomposition explains why the bias vector accounts for ∼85% of improvement over untrained baselines (Section 3.4): it provides a "default interpretation prior" that steers generation toward the space of valid feature descriptions, while the input vector specifies *which* description within that space.

## J. Zero Vector Interpretations

To directly observe what the bias vector encodes, we generate SelfIE outputs for the zero vector $h = \mathbf{0}$ across adapters trained on different datasets. Since $f(\mathbf{0}) = \alpha \cdot \mathbf{0} + b = b$ for scalar affine adapters (and $f(\mathbf{0}) = b$ also for full-rank), zero-vector outputs reveal the bias in isolation.

Table 17 shows greedy-decoded outputs. The key distinction is between SAE-trained and Wikipedia-trained adapters:

- **SAE-trained adapters** generate abstract, categorical descriptions. Sampling at temperature 1.0, Llama Scope produces "numerical values and measurements," "references to software development," "technical terms related to databases"... all generic feature categories (in lowercase). Goodfire samples frequently begin with "The assistant is..." (explaining, providing options, proposing alternatives), suggesting its training labels emphasize instruction-following contexts.

- **Wikipedia-trained adapters** generate specific encyclopedic topics. Llama-8B samples skew toward Russian culture (Turgenev, Soviet filmmakers, Russian émigré authors). Qwen-7B samples favor music (Beatles, Elton John, rockabilly, synthesizers). Qwen-14B samples favor science and history (Ebola outbreaks, dinosaur extinction, Darwin, particle accelerators).

The bias vectors have learned what a "generic" SAE feature label versus a "generic" Wikipedia topic description looks like.

*Table 17.* Zero-vector interpretations reveal each adapter's learned prior. SAE-trained adapters (top) generate abstract feature descriptions; Wikipedia-trained adapters (bottom) generate specific encyclopedic topics.

| Adapter (training data) | Greedy output ($h = \mathbf{0}$) |
|---|---|
| Llama-8B (Llama Scope) | "references to specific locations and their characteristics" |
| Llama-8B (Goodfire) | "The assistant is providing a list of options or alternatives" |
| Llama-8B (Wikipedia) | "the Russian composer who wrote the opera Eugene Onegin" |
| Qwen-7B (Wikipedia) | "the 1960s British rock band that pioneered psychedelic rock" |
| Qwen-14B (Wikipedia) | "the 1976 outbreak of the Ebola virus in Zaire" |

These results confirm that the bias vector learns a distributional prior over valid interpretations, while the input vector selects among them.

## K. Detection-Optimal vs. Generation-Optimal Labels

**Each row in Tables 2 and 3 uses different labels for different columns.** The two-phase select-then-rescore protocol of Appendix A.5 selects the best label per latent under whichever metric is being reported. We want to be explicit that this means each *column* of a given row uses its own per-latent winning label, not a shared one. The hit-rate column reports the result of (i) selecting the candidate label that maximized first-pass generation hit rate per latent, then (ii) freshly rescoring that label; the detection-F1 column reports the result of (i) selecting the candidate label that maximized

first-pass F1 per latent, then (ii) freshly rescoring that label. The two selected labels for a given latent are often *not* the same string.

We do not regard this as something to hide. On the contrary, part of what we are claiming is that a SelfIE adapter is more useful than a method that produces a single canonical label per vector. A single adapter run produces a small set of candidate labels per latent (across the scale grid in our setup), and which candidate is most useful depends on what the label is being used for. Generation scoring and detection scoring stress-test different aspects of a label, and the labels that do best on each tend to be different strings.

**What the two metrics measure.** The two SAE evaluation metrics ask different questions of a label:

- **Detection scoring** asks how reliably a label *distinguishes* activating from non-activating contexts. Given pre-compiled snippets that did or did not activate the latent, the judge model is shown the label and predicts the class of each snippet. F1 is high when the label admits the activating snippets and rejects the non-activating ones.

- **Generation scoring** asks how reliably a label *elicits* activating contexts. The label is given to a generator model with instructions to produce an example, and the latent is checked on the resulting text. Hit rate is high when the natural completion of the label tends to fall inside the latent's activation region.

**Core vs. periphery.** A useful intuition is that a typical SAE latent has a *core* of concepts that activate it strongly and prototypically, surrounded by a *periphery* of related concepts that activate it less reliably. The two metrics select labels that pick out different parts of this region:

- The detection-optimal label tends to outline a broader region that includes part of the periphery, because covering peripheral activators reduces false negatives without inflating false positives too much, and that tradeoff is what F1 rewards.

- The generation-optimal label tends to describe the core directly, because the surest way to make a generator's continuation activate the latent is to point it at the highest-probability activating concept rather than at the broader category that contains it.

The result is a systematic shift in label *specificity*: detection-optimal labels read like broad category descriptions, while generation-optimal labels read like specific concepts inside that category.

**Examples.** Table 18 shows the latents for which the detection-optimal and generation-optimal labels disagree most strongly, drawn from the candidate label sets produced by the two Goodfire 70B adapters reported in Table 3 (one SA, one SA+LR; each adapter's six candidates come from the per-adapter calibrated scale subgrid of Appendix A.5, which is 0.5–5.5 for SA and 0.8–8.9 for SA+LR). To pick these examples we ranked the 1,000 evaluation latents per adapter by the sum of the two cross-label metric gaps,

$$\Delta = \big(\mathrm{F1}(\ell_{\mathrm{det}}) - \mathrm{F1}(\ell_{\mathrm{gen}})\big) + \big(\mathrm{HR}(\ell_{\mathrm{gen}}) - \mathrm{HR}(\ell_{\mathrm{det}})\big),$$

where $\ell_{\mathrm{det}}$ and $\ell_{\mathrm{gen}}$ are the two labels selected for a given latent (by detection F1 and by generation hit rate respectively), $\mathrm{F1}(\cdot)$ is detection F1, and $\mathrm{HR}(\cdot)$ is generation hit rate. The within-label, on-metric values $\mathrm{F1}(\ell_{\mathrm{det}})$ and $\mathrm{HR}(\ell_{\mathrm{gen}})$ are the rescored scores that contribute to Tables 2 and 3; the off-metric values $\mathrm{F1}(\ell_{\mathrm{gen}})$ and $\mathrm{HR}(\ell_{\mathrm{det}})$ are first-pass scores of the same two labels evaluated under the other metric. $\Delta$ is large precisely when each label scores well on its own metric and poorly on the other—i.e., the two metrics "don't agree on which label is best." We report the top 5 latents by $\Delta$ for each adapter, with no further filtering.

The labels in Table 18 are reproduced verbatim from the adapter's output, with the exception of ellipses for brevity. The same-selection-criterion scores (bolded) are significantly larger than the scores using the metric the label was not selected on. The two metrics genuinely select different labels rather than slightly different orderings of similar labels.

The pattern is consistent across both adapters and across all ten latents shown. Detection picks broad categorical descriptions ("technical explanations," "code syntax and structure," "rhetorical questions and statements"); generation picks specific concepts that lie inside those categories (variable assignment in a particular language, Windows OS history, a specific pun). The off-metric values are uniformly near zero, confirming that these are not labels that "almost" tie under the other metric.

**The two metrics prefer opposite ends of the scale grid.** Looking at the top 20 latents by $\Delta$ for each adapter (the larger sample from which the 5 in Table 18 are drawn), the generation-optimal scale is strictly larger than the detection-optimal scale in 17 of 20 cases for *both* adapters (85% in each). Detection-optimal scales concentrate at the low end of the grid (median $s = 2.1$ for SA+LR, $s = 0.8$ for SA), while generation-optimal scales concentrate at the high end (median $s = 4.45$ for SA+LR, $s = 2.1$ for SA). This is consistent with the bias-vs-input-vector decomposition documented in Appendix I: at low scales the adapter's bias vector dominates and the output reflects a generic/categorical prior over feature labels (well-suited to outlining a broad activation region for detection), while at high scales the

*Table 18.* Top 5 latents by metric disagreement $\Delta$ (defined in Appendix K) for each of the two Goodfire 70B adapters from Table 3. For each latent, the top row is the detection-optimal label and the bottom row is the generation-optimal label, with the corresponding scale grid value $s$ shown next to each. Bold marks the metric used for selection (rescored under an independent seed); the unbolded off-metric value is the first-pass score of the same label. The top-row bold values for "Det. F1" are exactly the values that average into the corresponding column of Table 3, and likewise for the bottom-row bold values for "Gen. HR."

| Latent | Scale | Label | Det. F1 | Gen. HR |
|---|---|---|---|---|
| *SA+LR adapter (Goodfire 70B):* | | | | |
| #3890 | 0.8 | "Technical explanations of how software systems work" | **0.80** | 0.00 |
| | 8.9 | "embedded systems" or "operating systems", but in the context of software development, "embedded… | 0.00 | **1.00** |
| #8102 | 2.1 | "Formal or polite language, especially in requests or expressions of gratitude" | **0.75** | 0.00 |
| | 8.9 | "to condescend" or "to stoop," and it is often used in a phrase such as "to stoop to someone's level… | 0.00 | **1.00** |
| #3505 | 2.1 | "Someone's identity, abilities, or characteristics are being described or defined" | **0.71** | 0.00 |
| | 5.5 | "My existence or presence is…" | 0.00 | **1.00** |
| #9848 | 3.4 | "The person's name is being requested or provided" | **1.00** | 0.00 |
| | 1.3 | "The assistant is asking for the user's name" | 0.31 | **1.00** |
| #329 | 2.1 | "Numerical values and quantities" | **0.76** | 0.10 |
| | 1.3 | "The assistant is providing a numbered list or sequence" | 0.00 | **0.90** |
| *SA adapter (Goodfire 70B):* | | | | |
| #4635 | 0.8 | "Rhetorical questions and statements of disbelief or astonishment" | **1.00** | 0.00 |
| | 3.4 | "refers to a play on words, specifically a pun on the word "chicken" and "egg… | 0.00 | **1.00** |
| #2537 | 2.1 | "The assistant is referring to code examples in the conversation" | **0.95** | 0.00 |
| | 3.4 | "This is a reference to the current figure or plot in a matplotlib context, typically used in documentation or tutorials to refer… | 0.00 | **1.00** |
| #6342 | 0.5 | "Technical code documentation and explanatory text" | **0.86** | 0.00 |
| | 2.1 | "A CMake variable that holds the path to a library or executable" | 0.00 | **1.00** |
| #6603 | 2.1 | "Technical and historical explanations of software and technology" | **0.86** | 0.00 |
| | 1.3 | "Windows operating system release history and timeline" | 0.00 | **1.00** |
| #10243 | 0.5 | "Syntactic formatting patterns in programming languages" | **0.95** | 0.00 |
| | 1.3 | "SIMD instruction patterns in assembly code" | 0.17 | **1.00** |

input vector dominates and the output reflects the specific direction of the latent (well-suited to pointing a generator at the core concept).

**Looking up the latents.** Each of the Goodfire 70B SAE latents in Table 18 has a public Neuronpedia dashboard at https://www.neuronpedia.org/llama3.3-7 0b-it/50-resid-post-gf/3890 (with the latent index in place of 3890). Comparing each dashboard's activating contexts and top-logit tokens against the two labels in the table makes the core-vs.-periphery distinction more concrete than the labels alone can convey.

The fact that these divergences are systematic, rather than the two metrics simply ranking similar labels in slightly different orders, is itself evidence for the core/periphery picture. If both metrics were measuring noisy versions of a single underlying "label quality," the same string would tend to win both selection steps; instead, the two selection steps preferentially surface labels that occupy different parts of the latent's activation region, and these labels are reliably

found at opposite ends of the scale grid. We view this as a reason to prefer methods that produce a structured spread of labels per latent, like a SelfIE adapter swept across a scale grid, over methods that commit to a single canonical label.

