# OpenReview forum: "Learning Self-Interpretation from Interpretability Artifacts: Training Lightweight Adapters on Vector-Label Pairs"
_ICML.cc/2026/Conference — ICML 2026 regular_

### Official Review · Reviewer_pvHB · 2026-03-09

**Soundness:** 3
**Presentation:** 3
**Significance:** 3
**Originality:** 3
**Overall Recommendation:** 4
**Confidence:** 3

**Summary:**

This paper improves LLM self-interpretation by training lightweight adapters on existing interpretability artifacts, SAE decoder vectors paired with auto-interpretability labels and contrastive activation vectors paired with synthetic topic descriptions, while keeping the language model frozen. The central finding is that a very simple scalar affine adapter is often sufficient: its learned bias vector contributes most of the improvement, functioning as a default interpretation prior, while the input activation supplies the instance-specific semantics. Across experiments, trained adapters substantially improve topic identification from contrastive activation vectors, can outperform the original SAE training labels under generation-scoring evaluation, and generalize better across datasets and layers when the adapter remains simple.

**Compliance With Llm Reviewing Policy:**

Affirmed.

**Key Questions For Authors:**

1. Probing classifiers are a standard way to test whether information is linearly accessible in representations. Have the authors compared the bridge-entity results against linear probes? If a simple probe achieves similar detection performance, the adapter’s main added value may be in producing natural-language descriptions rather than uniquely surfacing otherwise inaccessible information.

2. A central headline claim is that trained adapters can outperform the original training labels, but this seems to depend on the evaluation metric: under generation scoring the trained adapters win, whereas under detection-style scoring the auto-interpretability labels are slightly stronger. Could you clarify what each metric is actually capturing, and which one is viewed as the more faithful measure of self-interpretation quality?

3. The paper’s main distinction from concurrent work is to freeze the language model and learn only a lightweight affine transformation, motivated by the claim that this best preserves “privileged access.” Have you run any direct comparison against a lightweight fine-tuning baseline such as LoRA, or any ablation of the injection setup, to test whether this design choice is empirically important rather than only conceptually motivated?

**Limitations:**

yes.

**Strengths And Weaknesses:**

**Strengths**

- The empirical analysis is well controlled. The architecture ablations are systematic, the Taboo baseline helps separate self-interpretation gains from general descriptive capability, and the controlled experiments distinguishing intrinsic dimensionality from label count show strong scientific care.
- The paper is clearly written and well organized. Figure 1 communicates the pipeline effectively, and the appendices are unusually thorough, including prompts, hyperparameters, compute details, scale-selection protocol, and training curves. The discussion of related and concurrent work is also clear.
- The paper offers a useful conceptual reframing by treating interpretability artifacts as supervision for self-interpretation. The finding that the bias vector accounts for most of the gain provides real mechanistic insight, and the bridge-entity experiment gives a concrete example of recovering latent intermediate reasoning without verbalized chain-of-thought, all while keeping the interpreted model frozen.

**Weaknesses**

- A central headline claim that trained adapters can outperform the original training labels—holds under generation scoring, but the paper’s own detection-scoring results still slightly favor the auto-interpretability labels. This difference is real and should be discussed more explicitly.
- The bridge-entity experiment is interesting, but it is evaluated only on a filtered subset of TwoHopFact and without comparison to simpler baselines such as linear probes, so it remains unclear how broadly this result extends to other forms of latent reasoning.
- The method depends on existing interpretability artifacts for supervision, and the empirical scope is still fairly narrow: the evaluations focus on contrastive topics, SAE features, and bridge entities.
- Relative to concurrent work such as LatentQA and related self-explanation papers, the paper’s main distinction is freezing the model and learning only a lightweight affine input transformation. That is a meaningful design choice, but the claimed advantage for preserving privileged access is argued conceptually rather than demonstrated in a direct head-to-head comparison.

---

> ### Author Rebuttal · Authors · 2026-03-31
>
> We thank Reviewer pvHB for the thorough review and for noting the scientific care in our experimental design.
>
> **Generation scoring vs. detection scoring.** Detection tests a label as a *recognizer* (can it distinguish activating from non-activating examples?); generation tests it as a *specification* (can it guide creation of new triggering instances?). These come apart: an overly broad label may recognize well but generate poorly, while a narrow label capturing one activation mode may generate well but miss others in detection. We will add a brief explanation of this intuitive difference to the camera-ready.
>
> Table 2 reports best-of-6 metrics for all methods, which is appropriate for the comparison it makes. However, a simpler and arguably more natural comparison is also possible: the practical workflow for trained SelfIE is to generate labels at multiple scales and select the best one, producing a *single label per latent*—exactly what auto-interp provides. We can then compare these two one-label-per-latent sets head-to-head. Concretely: we use the multi-scale protocol to select the highest-F1 SelfIE label per latent, then re-score that selected label set fresh (to avoid upward bias from selecting on the same noisy metric). Under this direct comparison, trained SelfIE achieves F1 = 0.725 ± 0.003 vs. 0.684 ± 0.003 for auto-interp labels, a significant improvement. The trained adapter thus outperforms auto-interp labels under both generation scoring (Table 2) and detection scoring when compared at the level of "one best label per latent."
>
> **Linear probe baseline.** We have run this experiment. Because the median bridge entity appears only once in the filtered dataset, a standard multi-class probe cannot generalize. We instead embedded the main textual labels of all entities in the TwoHopFact dataset and trained linear maps from last-token activations to GTE-Large embedding space using InfoNCE loss with optimized temperature, one probe per layer.
>
> Both methods produce per-position scores that must be aggregated to the prompt level. For SelfIE, we draw $n{=}10$ generations at each token position; if $c$ of these contain the bridge entity, $\text{pass}@k = 1 - \binom{n-c}{k}/\binom{n}{k}$. For the probe, each position yields a retrieval probability $p$, giving $\text{pass}@k = 1-(1-p)^k$. We then pool across token positions: $P = 1 - \prod_t(1 - \text{pass}@k_t)$, asking whether *any* position in the prompt successfully surfaces the bridge entity in $k$ attempts. Each method uses its single best layer:
>
> | $k$ | SelfIE (L19) | Probe (best L) | Sig. |
> |---|---|---|---|
> | 1 | 47.4% | **57.1%** (L28) | $p < 0.001$ |
> | 5 | 66.7% | 63.7% (L30) | n.s. |
> | 10 | **73.0%** | 67.0% (L30) | $p = 0.01$ |
>
> The probe has higher single-shot accuracy; SelfIE produces more diverse outputs and uncovers bridge entities more reliably with multiple attempts. When pooling over *both* tokens and layers (32 per-layer probes vs. one SelfIE adapter), SelfIE wins decisively at all $k$ (e.g., 91.0% vs. 85.6% at $k{=}10$, $p < 0.001$). SelfIE thus exceeds linear probe performance at the prompt level while producing free-form natural language rather than selecting from a fixed set.
>
> **LoRA comparison.** We trained Activation Oracles (LoRA rank 4) on identical data. Wikipedia topics: 81.6% ± 0.6% recall@1 (LoRA) vs. 82.9% ± 0.6% (SelfIE). Goodfire SAE generation scoring: 66.9% ± 0.6% (LoRA, 1 greedy + 5 samples at $T{=}0.7$) vs. 62.8% ± 0.6% (SelfIE, best-of-6 scales). Trained SelfIE achieves comparable but not superior raw performance. Its advantages are qualitative: the learned transformation is itself interpretable (Appendix I–J), and the frozen model avoids the label-drift problem noted by Li (2026, "Introspective Interpretability," belindal.github.io/introspection): *"Training the same LM to explain itself means that the underlying labels are constantly shifting—once you train a model to explain itself, its behavior might change to the point that the original explanations are no longer faithful."*
>
> **Narrow scope.** The training framework is general: any (vector, label) pairs serve as supervision, and anything concurrent methods train on, we can train on. We prioritized analytical depth over breadth within the constraints of a single paper.
>
> We kindly ask Reviewer pvHB to consider strengthening their support if these results address their concerns.

---

> > ### Author Rebuttal · Reviewer_pvHB · 2026-04-02
> >
> > Thanks for the rebuttal. It addresses several of my concerns. However, I consider the concern about the paper’s relatively narrow empirical scope to be only partially resolved. Therefore, I will keep my current score.

---

> > > ### Author Response · Authors · 2026-04-08
> > >
> > > We thank Reviewer pvHB for the constructive engagement and for acknowledging that several concerns were addressed.
> > > On empirical scope: we would note that the rebuttal introduced three new baselines not present in the original submission—LoRA-based Activation Oracles, linear probes for bridge entity extraction, and a head-to-head detection scoring comparison—which will all appear in the camera-ready.
> > >
> > > We also want to flag a methodological improvement we are making to the evaluation in Table 2, which we believe addresses the reviewer's metric-dependence concern. The submitted version used a best-of-6 protocol where each label source produced up to 6 candidate labels per latent, each candidate was scored independently, and the maximum score across candidates was aggregated to produce the overall number. This meant that even the auto-interp baseline (which has only one label per latent) was reported as "auto-interp ×6" with six identical candidates, an awkward formalism that obscured what was really a simpler comparison.
> > >
> > > For the camera-ready, we are switching to a cleaner two-stage procedure: (1) each label source generates up to 6 candidates, then selects 1 best label per latent based on a selection metric; (2) the selected labels are freshly rescored to avoid selection bias, and the overall score is simply the average across latents. Under this procedure, auto-interp labels pass straight through to final scoring (since there is only one label per latent to begin with), giving the most natural baseline possible. Trained SelfIE benefits from candidate diversity across scales, but is held to the same one-label-per-latent standard. Under this evaluation, trained SelfIE achieves F1 = 0.725 ± 0.003 vs. 0.684 ± 0.003 for auto-interp labels on detection scoring, showing that the "outperforms training labels" claim holds under both metrics, not just generation scoring. We will completely redo Table 2 with this procedure, which we believe is both simpler and fairer than the original.
> > >
> > > We are also revising the introduction and conclusion to better contextualize the method's novelty and the significance of the frozen-model property for the broader interpretability community.

---

### Official Review · Reviewer_GEn3 · 2026-03-12

**Soundness:** 3
**Presentation:** 2
**Significance:** 3
**Originality:** 3
**Overall Recommendation:** 4
**Confidence:** 2

**Summary:**

This work studies the use of LLM self-interpretability with trained adapters: the authors train adapters that transform a given LLM activation, where the transformed activations are injected as part of an interpretability prompt that is given to the same LLM and used to generate a textual description of the "meaning" of this activation. The paper includes multiple experiments of training on weakly-labeled data from existing SAE baselines, comparing the results when learning different transformation functions (scale, affine and low-rank parameterizations).

**Compliance With Llm Reviewing Policy:**

Affirmed.

**Final Justification:**

As I wrote in my review and comments, I believe the work makes a solid contribution. I am nevertheless ambivalent because I found the narrative and overall message confusing and the details a bit difficult to follow. While the authors have responded seriously in their rebuttal, I have not changed my evaluation since I cannot judge to what extent these presentation issues will be resolved in the text revision.

**Key Questions For Authors:**

1. If the adapter learns a scale $\alpha$ in training, how come we still need to modulate that with another scale? And from the other direction - if we in any case use the best-of-N scales based on the outcome, how is the "Scale-only" training in Table 1 different from "Identity"?
2. From Table 1 we reach the conclusion that SA+LR > SA, But then in Table 2 this does not necessarily seem to be the case. Can you explain what is the reason for this? Is it the use of different metrics (validation CE loss vs. generation-scoring)?
3. Do you think that the strong results for Wikipedia-based training in §3.5 can be partly attributed to the appearance of the relevant question entities in the training data?

**Limitations:**

See weakness #1 - the limitations of applying this method in a practical use case should be stated more explicitly

**Strengths And Weaknesses:**

Strengths
1. Quite extensive experimentation with different setups and plenty of interesting results that could be informative for the communities working on interpretability. Also some really interesting findings in Appendix I+J (perhaps these should be given more prominence in the paper).
2. In some cases we see that the results of training can even surpass the noisy labels that were used for training.
3. Keeping the base model frozen and the number of parameters limited is IMO a good approach for more trustworthy and generalizable (self-)interpretability methods.

Weaknesses
1. More than anything, I found the narrative somewhat confusing here. As a method paper that tries to demonstrate the value of the proposed technique, I am missing a clear recipe for how one is supposed apply this interpretability method in practice. In particular, the authors use best-of-N to derive the results reported here, where the metric is how well the results capture the latents given by a separate trained SAE. But I did not quite understand how this corresponds to a usage in practice, where the goal is to get from a model representation to a textual interpretation (How would the user choose the right scale? Would one need an SAE also for test-time, or just for the training? etc.). In contrast if the goal is more of an analysis paper then I do think the existing results are enough (we see the effects of different choices for the adapter parameterization, the generalization across training datasets, the importance of scale choice, the meaning captured by the bias vector and so on). But if that is the case, the narrative should be better framed to clarify that this is the main goal and contribution of the work, rather than proposing a "method" per se.
2. The main paper is not entirely self-contained, some parts are quite hard to understand without diving deep into the appendix. For example the use of scales is quite important to understanding the experiments but is only really explained in Appendix A.5; similarly the beginning of the discussion in §5 is very brief and not so clear on its own. Also in §3.1 I don't think it is stated explicitly enough what are the methods that produced the original "auto-interpretability" labels, and it is unclear which experimental results/methodology the "early stopping" mentions in page 4 relate to.
3. I didn't quite follow what we learn from the "Taboo" comparison in Figure 2 - all we see there is a ceiling effect, I am not sure what information this adds about scaling or about the method.

---

> ### Author Rebuttal · Authors · 2026-03-31
>
> We thank Reviewer GEn3 for the thoughtful review and for highlighting the Appendix I+J findings.
>
> **Practical recipe vs. analysis paper.** A fair characterization of a limitation. The paper is proof-of-principle work rather than an out-of-the-box tool, though we already use trained adapters internally to generate SAE labels for unrelated experiments, evidence it has some practical utility already. To address the reviewer's specific questions: no SAE is needed at test time, but only during training if SAE data is used; the Wikipedia contrastive vectors require only the model itself. In a practical feature-labeling pipeline, one would generate multiple candidate labels per vector and use an evaluation such as generation scoring to select the best. In this realistic workflow, varying the input scale is a valuable source of label diversity, qualitatively different from temperature sampling alone (best-of-6 scales achieves 62.8% ± 0.6% on Goodfire SAEs vs. 59.4% ± 0.6% for best-of-6 temperature samples at a fixed scale). As for choosing the right scale: trained adapters make this substantially more forgiving (Figure 7), reducing the problem from per-vector optimization to calibrating a single contiguous window of 6 scales from a logarithmic grid, once per (adapter, dataset) pair.
>
> **Self-containedness.** We will add a brief summary of the scale protocol in main text and clarify the "early stopping" reference on page 4. To answer the object-level question: §3.4 states that full-rank adapters overfit on SAE data and "early stopping doesn't fully mitigate this." This refers to selecting the minimum-validation-loss checkpoint during training: full-rank adapters begin overfitting partway through epoch 1 (Appendix B, Figure 3a), and their best checkpoint (val loss 1.691) still underperforms SA+LR's best checkpoint (1.619). The camera-ready will add an explicit cross-reference to Appendix B here.
>
> **Taboo baseline.** The Taboo comparison is essential for interpreting the scaling result. As §3.3 states: "Larger models generally perform better at nearly every task, so showing that self-interpretation improves with scale would prove little on its own." Taboo measures how much topic knowledge the model can express *directly* without self-interpretation. That this ceiling saturates at intermediate scale while SelfIE performance continues growing demonstrates that scaling gains come from improved self-interpretation specifically, not merely from knowing more topics. Without this control, we could not distinguish "self-interpretation works better at scale" from "bigger models know more."
>
> **Learned scale vs. external scale.** Learning a transformation that finds the optimal scale for any vector is attractive. We tried nonlinear direction-dependent scale functions (e.g., $f(h) = \exp(\mathbf{s} \cdot h + s_0) \cdot h/|h| + b$, and small neural networks), but these did not match best-of-N performance, likely because optimal per-direction scaling is too complex for simple parametric families. We thought adding these partially successful experiments to the paper would unnecessarily broaden the scope, and we leave solving the direction-dependent scale issue to future work.
>
> Regarding identity vs. scale-only in Table 1: "identity" means literal identity (injection at fixed scale 1.0, with no scale parameter in the adapter); "scale-only" optimizes a single scalar via gradient descent. The 0.29 loss improvement reflects the value of learning this scale. Table 1 reports validation loss with the as-trained parameters only, with no external scale search applied.
>
> **SA+LR vs. SA across tables.** The discrepancy indeed arises from different metrics. Cross-entropy loss (Table 1) measures fit to the specific label distribution. Generation scoring (Table 2) measures whether the label captures what the feature *actually does* in the model—a more semantically grounded metric. That SA sometimes outperforms SA+LR on generation scoring while losing on cross-entropy is a key finding: simpler adapters generalize beyond the training labels to ground-truth semantics.
>
> **Wikipedia entities in bridge evaluation.** Strong performance reflects distributional similarity between Wikipedia vital articles and TwoHopFact entities (both name real-world entities), not memorization of specific entities. Evidence: the adapter shows no train/val overfitting, confirming generalization to unseen topics (also see the "Qualitative example" in §3.2). So the difference in performance between the Wikipedia-trained adapter and the SAE-trained adapter (Appendix H) must not be due to the presence of specific entities in the dataset, but is likely due to the higher *task* similarity with labeling topics vs describing arbitrary linguistic features detected by SAE latents.
>
> We invite Reviewer GEn3 to raise any further concerns.

---

> > ### Author Rebuttal · Reviewer_GEn3 · 2026-04-01
> >
> > Thank you, I appreciate your clarifications and detailed responses.
> >
> > While the work to me does look sound and makes novel and extensive contributions, I did find the narrative and overall message confusing and and the details a bit difficult to follow, and from the comments of reviewer qYcB I see that I am not alone in this. Since I cannot judge to what extent this will be resolved in the text revision, I am keeping my evaluation unchanged.

---

> > > ### Author Response · Authors · 2026-04-08
> > >
> > > We thank Reviewer GEn3 for the candid assessment and for acknowledging the soundness and novelty of the contributions. For the camera-ready, we plan to restructure the introduction for a broader ML audience, leading with intuitive motivations before technical specifics. We will also foreground a clearer explanation of why frozen model weights are a desirable property for self-interpretation, and move key details (the scale protocol, early stopping clarification) into the main text so the paper is more self-contained.

---

### Official Review · Reviewer_qYcB · 2026-03-13

**Soundness:** 3
**Presentation:** 2
**Significance:** 3
**Originality:** 2
**Overall Recommendation:** 4
**Confidence:** 3

**Summary:**

In the area of LLM introspective interpretability, the authors propose learning lightweight adapters on interpretability artifacts (e.g., vector-label pairs from sparse autoencoder features). This solves the problem of hyperparameter sensitivity when training a model to inspect its own hidden states, instead relying on lightweight adapter models capable of making predictions from token embeddings. The results indicate that this is effective, and replicates or surpasses the interpretability benefits of baseline self-interpretability methods.

**Compliance With Llm Reviewing Policy:**

Affirmed.

**Final Justification:**

The paper presents a novel idea with significance to the interpretability community, and specific positive findings using that idea. I found the paper very hard to follow in terms of overall narrative, connection of results to high-level findings, and significance to the general audience. I raised my score from Weak Reject to Weak Accept after reviewing the rebuttal and the paper's experimental results, but still have concerns that the presentation of the work detracts significantly from the hard data in the paper. I would suggest to the authors to ask several people outside the subfield to read and grade the clarity and cohesiveness of the narrative. Both in comparison to prior published articles in this and other conferences, and to other papers reviewed in this session, it was hard to follow and understand how the individual experimental findings connected to the broader contribution.

**Key Questions For Authors:**

1. The core issue I would need resolved is understanding the clear impact of this paper and its differentiation from complementary works. As an audience member with some experience in mechanistic interpretability (while not being an expert), the introduction and background (or lack thereof) made it near impossible for me to evaluate the contribution. There does appear to be valuable work here, but if revisions were to be done, it would be to completely revamp the framing of the paper and situate the work in a very clear frame
2. A related recommendation on revisions to framing is to think more in general terms. As it stands, the paper is written for a mechanistic interpretability audience, particularly those with background knowledge in self-interpretability methods

**Limitations:**

The authors do not explicitly discuss the limitations of their paper, aside from a general disclaimer about interpretability practices' limitations.

**Strengths And Weaknesses:**

The overall significance and impact of the work is hard to assess, in large part due to the poor presentation (see below for details). There are myriad, conjoined analyses conducted, aligned by a general idea of training adapters on interpretability artifacts. These produce many datapoints about the effectiveness of such adapters, why they work, when they work, etc., but do not tie the work into a unified, key idea or method. The idea is evidently novel, though the impact of that novelty (even demonstrated through valid experiments) is limited, and hard to understand. The other concern with this paper is that this is too narrow of a contribution, having to do with one type of mechanistic interpretability practice and making a simple methodological step, which is to train on self-interpretability artifacts themselves. This is not a fundamental result that can be leveraged in other areas.

## Strengths

At each point, the experimental designs and decisions are well-justified (through reasoning and prior literature) and explained, and each result by itself can be understand. Overall, the experimental results are interesting, and indicate a positive result.

## Weaknesses

The presentation of the paper is overall poor, particularly in how the work is introduced and explained, and the lack of a narrative applied to the entirety of the paper. The introduction reads like we are half-way through a methodology section, and does little to contextualize the work within a broader context. This negatively affects our ability to assess the significance of the paper. In the Methods section, background information is assumed, and the methodologies of prior works (e.g., SelfIE and patching) are explained with a few sentences, with no introduction of formalisms or attaching them to intuitive concepts. This is written for a narrow field, not a broad audience.

In the results, it is difficult to tie them any experimental findings to one single, broad conclusion, beyond the fact that this (training adapters on interpretability artifacts) is an interesting, potentially valuable practice.

In evaluating the soundness of the paper, the implementation and experiments are valid, but it is relatively unclear to me what the principle claim is, and whether or not that claim is supported by the work. It appears it is, but I had a hard time tying the claims in the abstract to specific findings in the results. It's of course possible I am just missing them even after a few reads, but they are not readily accessible in the writing.

In terms of originality, it is unclear to me how fundamentally different this work is from those works cited as concurrent, complementary works in the Related Works section: "Concurrent work (Li et al., 2025; Karvonen et al., 2025; Huang et al., 2025) pursues a complementary approach: finetuning the LLM itself to answer questions about its activations ... Our lightweight adapters offer a different trade-off: fewer parameters, a frozen base model, and the ability to precisely characterize what the transformation learns." This seems to suggest that this work is merely a lighter weight version of similar work.

---

> ### Author Rebuttal · Authors · 2026-03-31
>
> We thank Reviewer qYcB for engaging with the work and identifying areas where clarity can be improved.
>
> **Presentation and narrative.** We acknowledge that the paper could better contextualize the work for readers less familiar with self-interpretation. We will revise the introduction to provide more intuitive grounding before technical details, and add background explaining Patchscopes-style activation injection at a conceptual level.
>
> **Unified contribution.** We respectfully believe the paper has a clear unified idea: *train an affine adapter on (vector, label) pairs to minimize label cross-entropy while keeping the LM frozen, then evaluate whether the resulting self-interpretations faithfully reflect internal semantics.* Multiple datasets and evaluations demonstrate that this single technique generalizes, strengthening rather than fragmenting the contribution.
>
> The reviewer noted difficulty tying abstract claims to results. The overall, most general claim is: *We show that training lightweight adapters on interpretability artifacts, while keeping the LM entirely frozen, yields reliable self-interpretation across tasks and model families.* The more specific claims supporting this vary in their quantifications of what "reliable" means and the choice of tasks:
>
> - *"Trained adapters outperform training labels (71% vs 63%)"*: Table 3 shows that on Llama-3.3-70B, a scalar affine adapter trained on Goodfire SAE labels generates new labels scoring 71.4% hit rate, surpassing both the original labels (54.8%) and their paraphrases (63.4%). The adapter learns a general vector→label mapping that outperforms the specific labels it trained on, showing reliable performance at SAE labeling.
> - *"94% recall@1 vs 1%"*: §3.2 shows the adapter accurately names topics encoded in activation vectors, and we know it's accurate because the correct topic is the #1 embedding search result 93.7% of the time vs. 1.3% for untrained SelfIE.
> - *"Decode bridge entities…surfacing implicit reasoning"*: §3.5 shows that on multi-hop questions where the model answers correctly with no chain-of-thought, the adapter extracts the never-verbalized bridge entity (e.g., "Plato" for "The author of The Republic was born in…→Athens") via exact substring match in 91% of cases. This supports our claim of "reliable self-interpretation" because we can use the trained adapter to interpret for us what the internal activations mean, uncovering the hidden bridge entity that we know ought to be represented somewhere.
> - *"Bias vector accounts for 85%"*: This is a specific technical finding in Table 1 that is surprising because alternative techniques e.g. Activation Oracles require training millions of parameters in order to be viable. We show that the scalar affine component with $d_{\text{model}}$ + 1 parameters is the heavy lifter, something which has no analogue in LoRA-based methods.
> - *"Simpler adapters generalize better"*: Table 2's cross-dataset columns show that scalar affine adapters trained on Wikipedia achieve 41–50% generation scoring on SAE features they never saw, while the higher-capacity full-rank adapter achieves only 23–35%. For trained self-interpretation methods like this to become truly powerful and reliable, one thing we're looking for is **cross-domain generalization of capabilities**, and in this case we show that the cross-domain transfer is actually higher when the architecture is more constrained. This finding has no analogue in LoRA-based methods.
>
> **Differentiation from concurrent work.** This distinction—freezing the model entirely vs. fine-tuning it—is not subtle; it is stated in the abstract, emphasized in the introduction, and discussed at length in Related Work and the Conclusion. The difference is also not one of parameter efficiency; compute costs are comparable since both approaches are dominated by base model inference. The distinction is qualitative: (1) the learned transformation is itself interpretable (the bias vector lives in token embedding space and can be decoded; Appendix I–J), with no analogue in LoRA methods; (2) the interpreter has *exactly* the same weights as the subject model, which no fine-tuning method preserves. This matters for future closed-loop experiments (§5). A direct LoRA comparison confirms comparable performance: Activation Oracles achieve 66.9% on Goodfire SAEs vs. 62.8% for SelfIE, and 81.6% vs. 82.9% on Wikipedia topics.
>
> **Scope.** We respectfully disagree that the contribution is too narrow. The method applies to any token-based language model with hidden activations, trains on artifacts already produced in abundance, and generalizes across three model families (Llama, Gemma, Qwen), two data sources (SAEs, contrastive vectors), and from monosemantic training to polysemantic inference.
>
> We invite Reviewer qYcB to identify remaining concerns that would increase their support.

---

> > ### Author Rebuttal · Reviewer_qYcB · 2026-04-02
> >
> > I appreciate the authors' response to my concerns, and indeed I think that they emphasis the technical contribution of the paper. Based on this and the other reviewers comments, I am comfortable raising my score from a 3 to a 4 due to the higher significance of the work within the interpretability literature.
> >
> > In terms of clarity and narrative, I agree with Reviewer GEn3 it is hard to assess how well a revision could address this. I maintain that it is hard to grasp the work at a high level, and that significant improvements to the narrative, flow, and organization of experiments and results would be necessary to be a high quality submission to a general audience conference. However, my rating of 1 on presentation was too harsh given the positive experiments and the communication of results of each experiment individually. I am raising my presentation score from 1 to 2, more in line with the consensus from other reviewers.

---

> > > ### Author Response · Authors · 2026-04-08
> > >
> > > We thank Reviewer qYcB for the increased scores and for engaging constructively with the technical contribution.
> > >
> > > We take the presentation concerns seriously and plan concrete revisions for the camera-ready. Specifically, we will restructure the introduction for a general ML audience less familiar with mechanistic interpretability, leading with intuitive motivations before technical details. We will also provide a clearer explanation of why frozen model weights are a desirable property for self-interpretation (currently underdeveloped in the submission), and move the best-of-N scale protocol description into the main text along with a concise explanation of why it is necessary. We believe these changes will substantially address the narrative and accessibility issues both Reviewer qYcB and Reviewer GEn3 identified.

---

### Official Review · Reviewer_PP3h · 2026-03-16

**Soundness:** 3
**Presentation:** 2
**Significance:** 3
**Originality:** 3
**Overall Recommendation:** 5
**Confidence:** 3

**Summary:**

The paper proposes to train adapters on top of frozen language models to improve self-interpretation, where the model itself explains its own internal activations. Instead of fine-tuning the model, the authors propose to learn a mapping from interpretability artifacts (SAE decoder vectors and contrastive activation vectors) to explanations keeping the model frozen. The experiments on the paper show that trained adapters produce more reliable interpretations than untrained SelfIE baselines and that can sometimes outperform the original auto-interpretability labels. The method generalizes across tasks and model sizes, and the authors find that simple architectures, such as scalar affine adapters, often work well and that interpretation quality improves with model scale.

**Compliance With Llm Reviewing Policy:**

Affirmed.

**Final Justification:**

Rebuttal Acknowledgement.

**Key Questions For Authors:**

- Have you considered a direct comparison with a LoRA adapter trained on the same vector-label pairs and the same evaluation protocol? Even a lightweight LoRA baseline (e.g., rank 4 or 8) would help quantify what is gained or lost by keeping the model frozen.
- What model was used to create the automated interpretability labels?
- Please define "hit rate" and "coverage" in the main body.

**Limitations:**

yes

**Strengths And Weaknesses:**

Strenghts

- The paper follows a timely line of work and provides a nice continuation of recent efforts on training models to self-explain their internal representations, instead of finetuning the model or training LoRA adapters, the authors freeze the model entirely and train a small adapter that "processes" the activations is a principled choice that sidesteps the concerns with model fine-tuning.
- The ALL-CAPS experiment and the zero-vector interpretations are interesting experiments to understand what the adapter's bias vector learns.
- The finding that trained adapters can surpass their own training labels is compelling.
- It is surprising that a scalar affine adapter with just d_model + 1 parameters is enough to achieve most of the gains, and that the bias vector alone drives ~85% of the improvement.

Weaknesses

- The paper motivates trained adapters as a solution to the scale sensitivity problem of untrained SelfIE, yet the trained adapters still rely on a best-of-6 protocol that searches over multiple external scale factors at inference time.
- The optimal adapter architecture varies across tasks, full-rank for contrastive vectors and scalar affine or SA+LR for SAE features, complicating its practical application.
- The injection is always performed at layer 0 (the embedding layer), but the paper provides no ablation on this choice. Given the finding that activations from different source layers can all be interpreted by a single adapter, it is natural to ask whether injecting the transformed activation at a different target layer would yield better or worse results.

---

> ### Author Rebuttal · Authors · 2026-03-31
>
> We thank Reviewer PP3h for the careful reading and positive assessment.
>
> **Need for input pre-scaling.** On Wikipedia contrastive vectors, the trained adapter works well at its as-trained scale: 82.9% recall@1 at scale 1.0 vs. 0.04% for untrained SelfIE, no scale search needed. For SAE features, scale 1.0 still substantially improves over untrained SelfIE, but the multi-scale protocol closes the remaining gap to the original auto-interp labels. The reason the adapter cannot learn a single universal scale is architectural: scalar affine adapters treat all directions identically and *cannot* adjust scale per-direction. This constraint aids cross-domain generalization (Table 2) but means that vectors requiring different injection magnitudes benefit from external modulation. Importantly, this is far simpler than untrained SelfIE, where Kharlapenko et al. found each vector requires individual scale optimization; here we calibrate a contiguous window of 6 scales from a larger logarithmic grid once per (adapter, dataset) pair, and that window works for all vectors.
>
> **Architecture varies across tasks.** This is true, and we view it as expected at this stage. We introduce adapter architecture as a tunable hyperparameter (a novel contribution relative to concurrent work). As training moves toward larger, multi-task datasets with Q&A framing (§5), we expect the optimal architecture to shift. Also, the variation itself is informative: it reveals that contrastive topic vectors and SAE decoder vectors occupy geometrically different regions of activation space (the former low-dimensional, the latter spanning nearly the full space; Figure 5). This diagnostic insight emerged precisely from systematically comparing architectures, and would not have been apparent otherwise.
>
> **Injection layer ablation.** We have run this experiment. On Llama-3.1-8B-Instruct with Goodfire SAE generation scoring, embedding-layer injection yields 62.8% hit rate, while injecting after the first transformer block gives 55.1% ± 0.7% and after the third gives 53.4% ± 0.7%. Embedding-layer injection is optimal: because the adapter output is optimized via gradient descent, it learns to place vectors in regions of embedding space the model processes well. Methods like Activation Oracles benefit from post-block injection because their trainable LoRA parameters reside inside those blocks.
>
> **LoRA baseline.** Since submission, we trained LoRA-based Activation Oracles (rank 4/8) on the same datasets with the same evaluation. On Wikipedia topics: 81.6% ± 0.6% recall@1 (LoRA) vs. 82.9% ± 0.6% (SelfIE). On Goodfire SAE generation scoring: 66.9% ± 0.6% (LoRA rank 4, 1 greedy + 5 temperature-0.7 samples) vs. 62.8% ± 0.6% (SelfIE, best-of-6 scales).
>
> We do not claim trained SelfIE outperforms LoRA overall. To be clear, the contribution is not parameter efficiency—the difference between 4,097 and ~10M trainable parameters is irrelevant in practice, since both are dwarfed by the 8B base model that dominates compute. Rather, trained SelfIE preserves exact weight identity between subject and interpreter, a property no other trained self-interpretation method shares. This matters for two reasons. First, it avoids the label-drift problem identified in Li (2026, "Introspective Interpretability," belindal.github.io/introspection): fine-tuning shifts model behavior so training labels may become unfaithful, requiring regeneration; our frozen-model approach is immune to this. Second, it opens a path toward closed-loop experiments where a model interacts with optimally transformed representations of its own activations, potentially observing or steering its own internal states, which requires that the model processing those representations be identical to the model that produced them. We discuss this direction under "RL from internal rewards" in §5.
>
> **Auto-interp labels.** Goodfire labels were generated by Claude 3.5 Sonnet. Llama Scope labels were generated by GPT-4o (He et al., 2024). The adapter's ability to surpass these labels (which the reviewer highlighted as compelling) holds regardless of which external model produced them, since the adapter learns from vector-label correlations rather than inheriting a quality ceiling from the labeling model.
>
> **Hit rate/coverage definitions.** Defined in the Table 2 caption; we will make the definitions more prominent in main text for the camera-ready.
>
> We kindly ask Reviewer PP3h, if these clarifications and new results are satisfactory, to consider strengthening their support for the paper.

---

> > ### Author Rebuttal · Reviewer_PP3h · 2026-04-03
> >
> > Thanks for the thoughtful response:
> >
> > > "trained SelfIE preserves exact weight identity between subject and interpreter, a property no other trained self-interpretation method shares"
> >
> > I would like to argue that, even though the model weights are kept untouched, the representation/feature itself is modified byt the adapter, which weakens the claim.
> >
> > > LoRA baseline. Since submission, we trained LoRA-based Activation Oracles (rank 4/8) on the same datasets with the same evaluation. On Wikipedia topics: 81.6% ± 0.6% recall@1 (LoRA) vs. 82.9% ± 0.6% (SelfIE). On Goodfire SAE generation scoring: 66.9% ± 0.6% (LoRA rank 4, 1 greedy + 5 temperature-0.7 samples) vs. 62.8% ± 0.6% (SelfIE, best-of-6 scales).
> >
> > Thanks for this, I understand you refer to "trained" SelfIE here.
> >
> > > Injection layer ablation.
> >
> > Thank you for running the ablation. However, I believe this does not fully address my original question. The experiment evaluates injecting an adapter trained for embedding-layer injection into later layers, which introduces a mismatch between training and inference distributions. This result is expected and does not establish that embedding-layer injection is optimal in general.
> >
> > Overall, I appreciate the additional experiments and clarifications provided in the rebuttal, particularly the LoRA comparison and the new ablation, which strengthen the empirical section. I will increase my score accordingly.

---

> > > ### Author Response · Authors · 2026-04-08
> > >
> > > We thank Reviewer PP3h for the increased score and thoughtful engagement throughout the rebuttal.
> > >
> > > Regarding the point that the adapter modifies the representation: this is true, but the modification is more analogous to soft prompting than to fine-tuning, as we note in Related Work. Fine-tuning can instill new behaviors of arbitrary complexity, while a learned input transformation, however powerful, can only shift the output distribution within the envelope of the model's existing learned computations, which are exactly preserved. We believe this is a meaningful distinction, though we appreciate the reviewer pushing us to articulate it more carefully.
> > >
> > > On the injection layer ablation: we apologize for being unclear. To clarify, we did not simply inject an embedding-layer-trained adapter at later layers. We trained additional adapters from scratch using layer 0 output and layer 2 output as the respective injection points, and evaluated each with its matched injection layer. The embedding-layer adapter still outperformed.
> > >
> > > We are grateful for the reviewer's support.

---

### Decision · Program_Chairs · 2026-04-30

**Decision:**

Accept (regular)

**Comment:**

This paper proposes training a lightweight adapter on interpretability artifacts (such as vector-label pairs) to improve the self-interpretation capabilities of Large Language Models. All reviewers gave a positive rating for the paper after the rebuttal.

Strengths:
1. **Interesting experimental results and compelling findings** (Reviewers PP3h, qYcB, GEn3, pvHB)
2. **Nice and comprehensive experiment design** (Reviewers PP3h, qYcB, GEn3, pvHB)
3. Reasonable approach (Reviewers GEn3)
4. Timely work (Reviewers PP3h)

Weakeness:
1. **Poor or confusing presentation** (Reviewers qYcB, GEn3)
2. Lack of ablation on layers (Reviewers PP3h)
3. Others like scale sensitivity, complicated practical application

Two reviewers convey concerns about the presentation and how much the authors could improve their presentation in the final version. I, myself, am worried about this, too. If accepted, I strongly suggest the authors to improve their narratives according to the reviewers' suggestions, which will further improve the impact of the paper and let a broader group of readers to appreciate its interesting findings.